# Interactions between Dislocations and Boundaries during Deformation

**DOI:** 10.3390/ma14041012

**Published:** 2021-02-21

**Authors:** Hongjiang Pan, Yue He, Xiaodan Zhang

**Affiliations:** 1Faculty of Materials Science and Engineering, Kunming University of Science and Technology, Kunming 650093, China; 2Department of Mechanical Engineering, Technical University of Denmark, 2800 Kongens Lyngby, Denmark; 3Department of Chemical and Materials Engineering, University of Auckland, Private Bag 92019, Auckland 1142, New Zealand; yue_he_1225@outlook.com

**Keywords:** dislocation–boundary interaction, dislocation–interface interaction, deformation twin-boundary interaction, size effect, boundary structure, boundary strengthening, characterization techniques

## Abstract

The interactions between dislocations (dislocations and deformation twins) and boundaries (grain boundaries, twin boundaries and phase interfaces) during deformation at ambient temperatures are reviewed with focuses on interaction behaviors, boundary resistances and energies during the interactions, transmission mechanisms, grain size effects and other primary influencing factors. The structure of boundaries, interactions between dislocations and boundaries in coarse-grained, ultrafine-grained and nano-grained metals during deformation at ambient temperatures are summarized, and the advantages and drawbacks of different in-situ techniques are briefly discussed based on experimental and simulation results. The latest studies as well as fundamental concepts are presented with the aim that this paper can serve as a reference in the interactions between dislocations and boundaries during deformation.

## 1. Introduction

The mechanical properties of metals at ambient temperatures (from about 0 to 200 °C [1,2,3]) are mainly affected by their microstructures, except for the intrinsic properties determined by their chemical compositions [4,5]. As plastic deformation of metals proceeds via the dislocation motion, including nucleation, multiplication and slip of full dislocations with an edge/screw/mixed characteristics as well as nucleation and propagation of partial dislocations/deformation twins [6,7,8,9]. Compared with the dislocation–dislocation interaction, interactions between dislocations and boundaries, such as grain boundaries, twin boundaries or phase interfaces, play a more important role in the engineering of the mechanical properties such as strength [10,11,12,13,14,15,16,17,18].

It is well known that the grain refinement by thermomechanical processing is an effective way for the improvement of strength. Generally, the boundary strengthening is based on the impediment of dislocation motions by all sorts of boundaries [16,18,19,20,21,22]. The concept of grain boundary engineering (GBE) has been developed based on the knowledge of the grain boundary structure and energy in the past twenty years [23,24,25]. These developments have motivated scholars to discover the nature of dislocation–boundary interactions at different length scales. For decades, the progress has been made in the studies of boundary structures experimentally [26,27], as well as in the field of interactions between dislocations and boundaries, such as the processes and products of interactions [12,13,14,16,21,28]. Different influencing factors have been found and several models on the interaction mechanisms have been established [11,13,29,30,31,32].

In general, the boundary is an obstacle to dislocation motion and twin propagation during the deformation of metals. Dislocations or twins may either transmit across boundaries or be blocked by boundaries. The transmitted dislocation may be a rotated or a changed dislocation of the original incident dislocation, or it may be a new dislocation induced by the stress concentration derived from the pile-ups of incident dislocations.

The resistance of boundaries to dislocation motion during the interactions can be divided into three aspects [33]:The long-range elastic stress field between the dislocations and the boundaries, i.e., the image force;The dislocation accommodations to the boundaries when impinging on boundaries;The resistance of the adjacent crystal due to different orientations, Bravais lattices and lattice parameters.

All sorts of dislocation–boundary interactions show some similar behaviors and characteristics. This is due to the similarity between the dislocation–boundary and the deformation twin-boundary interactions: the essence of a deformation twin-boundary interaction is the dislocations of the incident deformation twin boundary interacting with the obstacle boundary, where the dislocation–boundary interaction is one of the essential parts during the whole process. However, different dislocations and boundaries have different interaction features, including interaction behaviors, dominating factors and transmission mechanisms.

Taken the transmission behavior as an example, it is reasonable to assume that the transmission behavior during the dislocation–boundary interactions can only occur when the resolved shear stress reaches an appropriate level and the transmission process must follow the rule of minimizing boundary energy [29,34,35]. Besides these, there are many other factors, which have significant influences on the transmission behavior. These factors include the misorientation [14,16,19,32], the geometry [20,30,36], the energy barrier [37], the dislocation type [31], the dimension [21], the Bravais lattice [13,38,39], the stacking fault energy (SFE) [31]. In order to predict the transmission behaviors of dislocation–boundary interactions in metals with a coarse-grained microstructure, numerous transmission mechanisms have been proposed. Most of them are established based on a single factor regardless of the others. However, the transmission behavior is influenced by a combined effect of many factors and none of them can be considered to be absolutely independent. The accuracy of models varies from case to case and should be restricted by the specific conditions. For instance, according to the in-situ transmission electron microscopy (TEM) observations on dislocation–twin boundary interactions, even in the case of a direct transmission without any residual dislocation, an absorption process can be found before the occurrence of any dislocation transmission [40]. Moreover, a recent study shows that the strain rate plays a significant role in the dislocation–grain boundary interactions [17]: the dislocations can transmit across a high angle grain boundary at a low strain rate but cannot at a high strain rate. This is ascribed to the different absorption processes at low and high strain rates, respectively. These findings indicate that the strain rate is not a negligible factor during the dislocation–grain boundary interaction. Even in the case where the strain rate is fixed, the observation of dislocation-boundary interactions is also related to the specimen size and the characterization techniques. Thus, a comprehensive transmission mechanism that can be applied in any circumstances is hard to obtain.

The dislocation–boundary interaction behaviors are significantly determined by the grain size [18,41]. This is because the dislocations are subjected to different stress states of image forces in metals with different grain sizes at micro- and nano-scales. For ultrafine-grained or nano-grained metals, a dislocation may be subjected to almost equivalent image forces derived from several boundaries due to the small grain boundary spacing. In this case, the resistance against dislocation motion in nano-grained metals is stronger than the one in coarse-grained metals, which is verified by a larger dislocation curvature in the grain interior [18,42,43]. The strong image force also changes the dislocation–boundary interaction behaviors in nano-grained metals through the dislocation behaviors: the dislocations nucleated initially at boundaries are stored and constrained in the vicinities of boundaries rather than glide into the grain interior. The vicinity of the boundaries is believed to be the favored places for plastic deformation. This phenomenon is different from that in a coarse grain, where the plastic deformation occurs throughout the grain with the slip of dislocations mostly in the grain interior. Moreover, the dislocation–boundary interactions change the boundary structure and generate severe distortions [41,44,45]. In the case of the deformation–twin boundary interaction, the tip of deformation twin subjects to a strong image force when approaching the boundary [46]. This makes the straight coherent twin boundary distorted and incoherent. Thus, the deformation twin is unable to grow larger and unlikely to transmit across the boundary.

For metals with a micro-scale boundary spacing, when a dislocation approaches a boundary, the image force derived from this boundary can be considered to be the dominant contributor of the resistance. In this case, the characteristics of the dislocations and boundaries, such as the types of dislocations and boundaries, geometrical conditions and energies during the interactions, are significant factors to the interaction between dislocations and boundaries. Besides, for the different types of dislocation–boundary interactions, the dominant factor may change. For example, the dominant factors of a dislocation-coherent twin boundary interaction are the dislocation types, the geometrical conditions and the SFE owing to the definite structure of coherent twin boundary [31,39,40]. However, for the twin–twin interaction, the Bravais lattice and the chemical composition are the primary factors due to the different twin boundary structures, slip systems and twin variants [13,38,39].

In the case of the dislocation–phase interface interaction, the phase interface is supposed to be a stronger barrier to dislocation motion compared with the grain boundary and twin boundary [21,30,47,48]. This is because the dislocation transmission across a phase interface must not only overcome the image force and energy barrier of the boundary, but also accommodate the different Bravais lattice, orientation and lattice parameters of the other phase. Thus, the primary factors of a dislocation–phase interface interaction are different from other sorts of dislocation–boundary interactions. The atomic bonding seems to be the most important factor: the resistance of a non-metallic compound interface is much stronger than the one between two different metallic phases. Additionally, according to the phase shape, size and interface configurations, such as particles and lamellae, their interactions between dislocations and interfaces are different. In terms of the non-metallic compound interface, the size of particle and lamella is the key factor [21,22]:The dislocations can only transmit across the non-metallic compound particle interface when the particle size is smaller than a critical value, which is always several nanometers;The slip are able to transmit across the non-metallic compound lamellar interface by slip steps only if the lamellar thickness is fine enough, otherwise the fracture of lamellae may occur due to the stress concentration of dislocation pile-ups.

With regard to the interfaces between two different metallic phases (particles or lamellae at the micro-scale), the alignment or deviation of slip systems of different phases is considered as a remarkable factor to predict the dislocation transmission behavior [47,48,49]. These indicate that the characteristics of two different phases, the morphology and dimension of phase interfaces play a more important role in this type of interaction.

In recent years, new discoveries on interactions between dislocations and boundaries are mostly obtained by in-situ scanning electron microscopy (SEM)/TEM mechanical testing. For example, at the nano-/atomic-scale, the detailed processes and products of interactions are investigated by TEM. Multiple phenomena are capable to observe, including the nucleation, slip, pile-up (dissociation or absorption), transmission (deflection or emission) and reflection of individual dislocations [16,33,50], the formation of steps and facets [12], the generation of local distortions and reconstructions [39] in the boundaries. While the macroscopic mechanical properties can be interpreted by the studies conducted at or above the micro-scale. For instance, to investigate the effect of grain boundaries and twin boundaries on slip traces and mechanical properties, a typical method is to perform in-situ SEM investigations on bi-crystal pillars [15,17,20,31,51]. Since the studies at different length scales have large distinctions in experimental conditions, specimen dimensions, deformation strains and strain rates, resolutions and research interests, each approach has achieved breakthroughs in its own specialized fields. Therefore, it is important and necessary to verify the analysis and conclusion by methods at different length scales to avoid one-sided understanding.

In this paper, based on experimental and simulation results, the structure of boundaries (the grain boundary, the dislocation boundary, the twin boundary and the multi-phase interface), the interactions between dislocations and boundaries in coarse-grained, ultrafine-grained and nanocrystalline metals (dislocation–grain boundary interactions, dislocation–twin boundary interactions, dislocation–phase interface interactions, deformation twin-grain boundary interactions and deformation twin–twin boundary interactions), and the available characterization methods are briefly reviewed and discussed as follows:Summary of boundary structure.Transmission mechanisms of dislocation–boundary interactions in coarse-grained metals.Interaction behaviors and influencing factors between dislocations and boundaries in coarse-grained metals.Effect of grain size: dislocation-boundary interaction behaviors and influencing factors in ultrafine-grained and nano-grained metals.Applications and characterization techniques.Summary and outlook.

## 2. Structure of Boundary

A boundary is a broad concept referring to the structure which separates crystals with different orientations (grain boundary or twin boundary), Bravais lattices and compositions (phase interface) [26]. During plastic deformation of metals at ambient temperatures, dislocations or deformation twins nucleate due to applied stresses. To a great extent, their interactions with the existing grain boundaries, twin boundaries and phase interfaces determine the mechanical properties of metals. Since the intrinsic structure of boundaries has a remarkable effect on these interactions, it is indispensable to understand the different types of boundaries before any specific narration.

Due to the thorough consideration of structure, composition and interfacial energy, it is reasonable to classify the boundaries in metals into two primary categories. The first category is that two crystals on either side of a boundary have the same phase but only differ by crystallographic orientations. This category includes the grain boundary, the twin boundary and the stacking fault. On the contrary, the other category is an interface that differentiates two crystals by composition or Bravais lattice. Generally, the complexity level and interfacial energy of a boundary have similar variation tendency. Both increase with the increase of misorientation, defect density, heterogeneity of Bravais lattices and composition across the boundary. Therefore, twin boundaries and grain boundaries with perfect CSL have simple and definite structures and relatively low interfacial energies. While crystals with poor matching and multiple defects at boundaries may generate large lattice distortions, which may form high interfacial-energy boundaries with complex geometries.

Hereby, the structures of grain boundaries and twin boundaries introduced by thermomechanical processing, as well as the interfaces formed in the matrix of alloy are briefly outlined in the following sections. To simplify the description, the word “boundary” only represents grain boundary and twin boundary. While the word “interface” in the following chapters mainly refers to the phase interface formed in the matrix of metals.

### 2.1. Grain Boundary

The structures of grain boundaries depend on the crystal Bravais lattices, interatomic interactions, point defects and segregations [52]. Although the real grain boundaries do not extend infinitely and may contain defects and facets, we only reviewed the grain boundary structures without any segregations in this paper.

The grain boundary is only about two or three atom layers in thickness but accommodates crystals with the same phase but different orientations by dislocations. The simplest grain boundary structure is a straight and non-defective low angle grain boundary: a periodic array of dislocations [53,54,55], as sketched in Figure 1. Notably, the low angle grain boundary may be constructed by two types of dislocation series (each series terminates atomic planes at one side of the boundary) [56,57]. For the high angle grain boundary, the structure may become more complicated.

Generally, there are five degrees of freedom for grain boundaries. The adjacent crystals can either rotate around three perpendicular axes to form a misorientation, or the grain boundary plane can rotate around two vertical axes with respect to the crystals [58]. However, the structures of grain boundaries have some regular patterns determined by minimizing the grain boundary energy. For example, the grain boundary plane is always the densest-packed plane and exhibits a good fitting between crystals [59,60,61,62]. In most cases, when two lattices form a grain boundary, one lattice can be obtained by rotating another lattice around a certain axis with an angle.

Grain boundaries can be roughly classified into symmetrical and asymmetrical boundaries according to their grain boundary planes and misorientations. If two crystals share an equivalent crystallographic plane at the grain boundary and show a mirror symmetry on each side of the boundary, it is a symmetrical boundary and vice versa [52]. The symmetrical tilt grain boundary is always used to interpret low angle tilt grain boundaries [63]. For the high angle tilt grain boundaries, this model only works well when the dislocations are aligned in an array with a constant spacing in the grain boundary [64].

The twist grain boundary is a type of asymmetrical grain boundary, where two crystals rotate around the common grain boundary plane normally and form a grain boundary, as shown in Figure 2. In the right illustration, the twist boundary is parallel to the plane of the figure, and one lattice is indicated by circles and the other by crosses. It can be seen that the coincident positions of circles and crosses are able to constitute a superstructure.

For the analytical quantification of grain boundaries with all tilt angles or orientations, the coincidence site lattice (CSL) and displacement shift complete lattice (DSC) models have been developed from the concept of coincident positions [65]. If two lattices with different orientations are overlapped with each other, some lattice points can be found to coincide periodically. The coincident points define a superstructure: a coincidence site lattice. The number of lattice points in the unit cell of a CSL is expressed as Σ, which characterizes the unit cell volume of CSL lattice compared with the crystal unit cell. For example, Figure 3 shows the Σ = 5 grain boundary and its CSL lattice. The Σ value is always odd and there are two special boundaries among them. The Σ = 1 boundary represents a perfect crystal without a grain boundary. Additionally, a twin boundary can be expressed as an Σ = 3 boundary in the CSL model.

The DSC model is developed from the CSL model and provides a more specific way to quantify grain boundaries. As shown in Figure 3, the CSL lattice can be retained by shifting a large displacement (CSL vector) to another coincidence point. However, there is another effective way to retain the CSL lattice: just preserving the coincidence instead of shifting a large displacement. When two lattices are overlapped with each other, a coarsest sub-mesh can be established. If all lattice points of one crystal shift a displacement of this sub-mesh vector along any direction, the coincidence still exists although the coincide points have changed. This sub-mesh defines a sub-lattice called the DSC lattice.

As mentioned above, a low angle tilt boundary consists of a periodic array of dislocations. The translation vectors of the DSC lattice are possible Burgers vectors for such grain boundary dislocations [66]. In other words, the DSC lattice dislocation is a sort of grain boundary dislocation. A low angle tilt grain boundary can be treated as small deviation from the Σ = 1 grain boundary (perfect single crystal). Geometrically, any practical grain boundary can be regarded as a small deviation from the perfect CSL lattice.

It is reasonable to assume that the grain boundaries with perfect CSL and low Σ are expected to be low interfacial-energy boundaries due to the high atomic density of the planes. The experimental results of symmetrical tilt boundaries validate the above assumption [26]. Figure 4 shows the dependence of the experimental grain boundary energies on the misorientation angle for the <110> symmetrical tilt boundaries in face-centered cubic (fcc) metals (Cu and Au). The visible drops of {111} and {113} planes are associated with the low-energy twin boundaries. It indicates that the grain boundaries with perfect CSL and low-indexed boundary planes (such as {111}, {100}, {110} planes) exhibit low energies. On the contrary, the other grain boundaries deviating from these special orientations show much higher boundary energies. These grain boundaries possess lattice distortion due to the elastic stresses between atoms or defects such as dislocations, facets and ledges at the grain boundaries.

Figure 4 demonstrates a clear correlation between the grain boundary energy and the atomic arrangement at the grain boundary. The actual grain boundary structure is more complicated than the above models. Figure 5 illustrates the lattice points of grain boundaries with perfect Σ = 5, 17, 37 and 1. The Σ = 17 and 37 grain boundaries can be regarded as an ordered sequence interspersed by Σ = 5 and 1 grain boundaries, i.e., a mixture of basic structural units labeled A and B. In most cases, it has been clarified that a boundary structure can be described as a periodic array of structural units. In this case, the interfacial energies of grain boundaries are expected to be lower than those predicted by the CSL model. According to this theory, a local elastic distortion region exists at a boundary deviating from the perfect CSL model, i.e., the unrelaxed type. This leads to the crystal planes in the vicinity of the boundary have larger spacing than the bulk value [67]. This phenomenon is termed as the volume expansion [68,69]. The volume expansion has been manifested by experiments in dominating the grain boundary energy for all types of grain boundaries in fcc and body-centered cubic (bcc) metals, which shows a proportional relationship between them [70].

### 2.2. Dislocation Boundary Introduced during Deformation

Dislocation boundaries form by dislocation accumulation during deformation of fcc and bcc single crystals and polycrystalline metals (e.g., Cu, Al, Ni and Fe), which is also called the grain subdivision. These dislocation boundaries separate relatively clean regions with crystallographic misorientation angles [71]. It is commonly observed in medium to high SFE metals, which are dominated by dislocation slip during plastic deformation at ambient temperatures such as cold rolling, torsion, drawing, etc.

During deformation, the dislocation slip follow certain active slip patterns, and tend to accumulate and form morphologies aligned as lines or boundaries, leaving other regions of the matrix with a relatively low density of dislocations. These dislocation boundaries divide the matrix into small cell-like regions with different orientations called “cell blocks”. As shown in Figure 6, two types of dislocation boundaries are distinguished by morphologies, which are called “extended boundaries” and “cell boundaries”. These two types of dislocation boundaries come into being by different slip mechanisms. The cell boundaries are formed mostly by mutual trapping of dislocations, which are termed as “incidental dislocation boundaries (IDBs)”. Whereas for extended boundaries, they are formed because the interaction of dislocations originated from different active slip patterns, which are termed as “geometrically necessary boundaries (GNBs)” [72,73,74].

The formation of different dislocation boundaries depends on the grain crystallographic orientations and deformation methods, and can be divided into three types [75,76]. The first type is totally composed of IDBs without GNBs and shows a cell structure, which forms in the grains with <001> parallel to the tensile direction. The other two types are cell-block structures and are composed of IDBs and GNBs. Among them, one type has straight and parallel GNBs and aligned approximately with the slip planes, and the other type has two sets of GNBs and deviate substantially from the slip planes. Figure 7 shows the TEM images of a GNB aligned approximately with the slip planes, which indicates that the nature of a dislocation boundary is a dislocation network due to the dislocation interactions [77,78].

Since the IDBs and GNBs have distinguished morphologies and are generated by different mechanisms, their average misorientation angles and average boundary spacing during deformation are remarkably different [79], as shown in Figure 8. The variations of average misorientation angles and average boundary spacing of GNBs change much faster than those of IDBs with the increase of strain.

When the deformation strain increases, both the morphologies and dislocation boundary parameters of metals evolve with it. In most cases, at low to medium strains, GNBs of metals are paralleled along certain planes, which may be a slip plane or a plane related to slip planes [79]. At relatively high strains (e.g., a true strain of 5.0), dislocation boundaries appear to be lamellar-structured and paralleled to the rolling plane [80]. With further increase of strain, the boundary spacing of both GNBs and IDBs decreases and misorientation angles across boundaries increase. In the case of severe plastic deformation (SPD), which introduced much higher strains than common production methods, further structural refinements can be achieved [71].

### 2.3. Twin Boundary

The interface between twin and matrix can be distinguished as coherent and incoherent ones, as shown in Figure 9a. The two straight parallel sides of twin 1 and 2 are coherent twin boundaries, which have low interfacial free energy. On the other hand, the short end boundary of twin 2 is an incoherent boundary, which has high misfit and high boundary free energy. The measured boundary free energy of incoherent twin boundaries in Cu is about 25 times as high as the value of a coherent one [58].

The twin boundary plane has atoms aligned along a coherent boundary (twin plane) with mirror symmetry. The atoms of twin plane belong to both lattices of two crystals without any lattice misfit. As mentioned in Section 2.1, a twin boundary can be regarded as a perfect Σ = 3 boundary with low interfacial energy. Figure 9b,c shows the illustration and the high-resolution transmission electron microscopy (HREM) image of a coherent twin boundary from the view of [110] direction in a fcc metal, whose twin boundary plane is the (111) close-packed plane and atoms show mirror symmetry arrangement across the boundary [27].

For metals with different Bravais lattices [81], they have different twin planes and growing directions. The twin plane and growing direction are {112} and <111> in bcc metals, {111} and <112> for fcc metals, and {1012} and <1011> for hexagonal close-packed (hcp) metals, respectively.

Twinning may occur during annealing or deformation, which are called “annealing twins” or “deformation twins”, respectively. Generally speaking, metals with high SFEs have a little chance to form annealing twins due to the close relationship between twins and stacking faults, and annealing twins tend to occur in metals with low SFEs. Thus, it is easy to find more annealing twins in brass than in pure copper due to a relatively lower SFE [58].

The formation of a deformation twin requires relatively high stress compared with dislocation slip because a twin is a planar defect while a dislocation is a line defect [81]. It requires not only the activation of certain dislocations, but also additional surface energy to form an interface. Therefore, the dislocation slip is always the dominating mechanism during deformation at ambient temperatures. Whereas twin occurs in some certain conditions including low deformation temperatures, specific orientations, a limited number of slip systems and high strain rates—where a high critical shear stress can be obtained. For example, in hcp metals, the favorite deformation twin is due to the limited number of slip systems. For bcc and fcc metals, which has various slip systems at ambient temperatures, the deformation twin is likely to form at low deformation temperatures and high strain rates.

The formation of a twin is closely related to stacking faults. For example, the fcc single crystal has a stacking order of ABCABC… along {111} planes. If the stacking order from a certain plane is reversed, which turn to be ABCACBACBA…, the two parts of the crystal form a mirror-symmetrical structure, i.e., a twin. Since the nature of twin boundary is a stacking fault, its interfacial energy is expected to be relatively low.

Generally, deformation twins are considered to form by the successive slip of Shockley partial dislocations. In fcc metals, there are three equivalent Shockley partials on {111} slip planes, which are b1 = a/6[211], b2 = a/6[121], b3 = a/6[112]. In fcc single crystals, the stacking sequence of atoms in successive close-packed planes can be demonstrated as ABCABCABCABC. If a partial dislocation glides along the slip plane, a whole layer of atoms will move along the slip plane and generate a stacking fault. Indeed, the slip of any partial dislocation may lead to the same shift in stacking sequence, i.e., A → B, B → C, C → A, although the three Burgers vectors have different orientations. This microstructural transition by the stacking fault mechanism can also be found in the fcc-hcp martensitic transition of rare-gas solids [82,83].

Figure 10 illustrates the step-by-step formation process of a deformation twin by the slip of partial dislocations on successive slip planes. Since the slip of partial dislocation results in the same change in stacking positions, the twin can be obtained by the slip of a series of partial dislocations with the same Burgers vector (Figure 10a) or different Burgers vectors (Figure 10b). These two approaches bring about different macroscopic strains in the long range although both of them form the same stacking sequence. The partials with the identical Burgers vector generate shear strains in the same direction, which may produce a relatively large macroscopic strain. This type of twins often exists in the matrix of coarse grains, whose morphologies are plates with straight twin boundaries. In contrast, the second approach may not form a large macroscopic strain since their shear directions vary with each other. This situation is common in nano-grained fcc metals, which shows no obvious plate-like morphology.

### 2.4. Interface between Different Phases

The interface seeks to maximize atomic matching and to minimize elastic strain for the minimization of interfacial free energy. As a result, the atoms of crystals at both sides of an interface tend to align along the close-packed planes or close-packed directions when an interface is formed [26]. Although the lowest interfacial free energy is achieved at the interface, a perfect matching is not always reached. Therefore, the interfacial energy increases with the increase of defect densities such as misfits, steps and ledges and the discrepancy of atomic bonding at interfaces.

In general, interfaces can be divided into three types according to the degree of atomic matching. “Coherency” is the terminology to describe the extent, and these three types are fully coherent, partly coherent and incoherent interfaces [84].

Fully coherent interface

This type of interface has complete matching between the atoms of two crystals and definite planes along the interface. It means that the lattice is continuous across the interface without disconnection or mismatch. A fully coherent interface is commonly observed between two phases with similar chemical composition but different Bravais lattice [85] (phase transformation) or two phases with both similar atoms size and Bravais lattice but different composition [86].

b.Partly coherent interface

Partly coherent interface refers to an interface with a similar arrangement of atoms between two crystal structures, which are accommodated by periodic misfit dislocations and/or steps at the interface. This probably happens when the two crystals have the same Bravais lattice, but different compositions and lattice parameters.

c.Incoherent interface

Incoherent interface means the atoms on each side of the interface show poor matching with each other. When the crystals have large differences in lattice parameters or the atomic bonding (such as covalent and metallic bonding), they seem to form an incoherent interface. Although it shows a tendency of aligning the close-packed planes of two crystals along the interface plane in order to minimize the interfacial energy, the interface is sometimes a high-indexed plane.

An incoherent interface with an almost flat interface plane between the Zr matrix and the ZrN precipitate in a Zr-N alloy is shown in Figure 11. It shows the faceted morphology along the interface and its orientation relationship is [450]_Zr/_/[101]_ZrN_ and (002)_Zr/_/(131)_ZrN_, a low-indexed Zr plane adjacent to a high-indexed ZrN plane [87]. However, the incoherent interface is not necessarily atomically flat. In this case, the incoherent interface shows a serrated shape and is composed of steps, which is called “disconnections” [88,89,90]. Although the crystals at both sides of a certain step may share a close-packed plane (also a low-indexed crystal plane), the whole interface may lie parallel to any arbitrary crystal plane according to different densities of steps.

## 3. Interactions between Dislocation and Boundary

Generally, the boundary is, more or less, a barrier for dislocation motion. Even in the weakest situation, i.e., a direct transmission across a boundary without any pile-up or residual dislocation at the boundary, the dislocations may be trapped in the boundary prior to being released from it, and the slip direction shall be changed [29,40].

The resistance of grain boundaries and twin boundaries mainly comes from the following three aspects [33]:When dislocations move towards the boundary, there exists a force against dislocation motion derived from the long-range elastic stress field between dislocations and the boundary, which is called the image force. The image force makes the group of following dislocations piled up in front of a boundary, and may increase due to the interaction between the leading dislocation and the boundary;Once impinging on the boundary, the incoming dislocation will face another resistance originated from the interaction between the dislocation and the boundary;When the dislocation is about to emit into the adjacent grain, the dislocation has to accommodate its orientation and Bravais lattice. This situation is more complicated for phase interface, whose resistance is stronger than those of grain boundaries and twin boundaries due to the dislocation accommodation to a different phase.

The dislocations may rotate, dissociate and reconstruct once impinging on the boundary. In some proper conditions, dislocation can overcome the impediment and transmit across the boundary. The dislocation transmissions of all sorts of boundaries are the result of stress concentration at boundaries induced by the accumulation of dislocations due to the increased applied stress. The transmission of dislocations across boundaries may dissociate, leave partial residual dislocations or form steps on the boundaries. Each type of dislocation–boundary interactions can be classified based on the different interaction behaviors, the dominating factors and the transmission mechanisms.

### 3.1. Dislocation–Grain Boundary Interactions in Coarse-Grained Metals

#### 3.1.1. Basic

As the most common case, the dislocation–grain boundary interaction in coarse-grained metals has been investigated in relatively comprehensive and intensive studies for decades as detailed in the following. Numerous studies have reported on this interaction characterized by static and dynamic in-situ TEM as well as SEM methods, and obtained lots of quantitative and qualitative results [15,16,17,19,20,29,33,34,35,37,50,91]. It has been found that the dislocation–grain boundary interactions of bcc, fcc and hcp metals are similar in most aspects [29]. However, it is still not fully interpreted.

When a dislocation impinges on a grain boundary, the dislocation lines may rotate to be parallel to the grain boundary owing to the image force. Then, the dislocations may dissociate or glide along the grain boundary once impinging on the boundary, which further appears to be absorption, transmission or reflection [29,33,34,40,50,92,93], as shown in Figure 12. The above situations can happen at the same time, i.e., a primary impeded dislocation may dissociate, transmit and reflect at the grain boundary, as shown in Figure 13.

In specific, a dislocation may maintain itself or dissociate into several dislocation partials with smaller magnitudes of Burgers vectors when it is impeded by a grain boundary. Some decomposed dislocations can glide along the grain boundary called “glissile dislocations”, while others reorganize with the local grain boundary dislocations to be a new periodic arrangement called “sessile dislocations” [33]. The glissile products can glide along the grain boundary and be impeded at steps or triple junctions, which may form cavitation due to the accumulated stresses. The sessile dislocation may change the local structure of the grain boundary. The absorption of dislocation is potential to emit dislocations with the increase of applied stress, which may transform into transmission or reflection.

Dislocation transmission can be considered as two types: direct and indirect. Direct transmission can be further divided into two modes. The first one refers to the situation that the slip planes of incoming and outgoing dislocation share the same line intersected on the grain boundary plane. This always happens when the boundary is a low angle symmetrical grain boundary sharing the same slip plane, so that the dislocations can glide across the boundary along the same slip plane by rotating a small angle deviated from the original slip direction. In this case, the Burgers vectors of incoming and outgoing dislocations do not change after crossing the grain boundary, and the energy required for transmission is equivalent to the cross-slip. Another type of direct transmission generates a residual dislocation at the grain boundary and the transmission energy is the formation energy of this residual dislocation. The magnitude $\left|b_{\mathrm{residual}}\right|$ of the residual dislocation can be expressed as [34,94]:***b***_residual_ = ***b***_in_ − ***b***_out_(1)

A criterion is found to be true in this mode that the magnitude of $\left|b_{\mathrm{residual}}\right|$ should be minimized [40,93].

Indirect transmission has the same cause as reflection. Both of them occur when the dislocations are piled up at the grain boundary. The increasing stress concentration may activate dislocation sources at the vicinity of the intersection point, which emits dislocations to the other grain or back to the inner part of the same grain, i.e., indirect transmission or reflection.

Besides the dislocation transmission, the interaction between a dislocation and a grain boundary may result in the formation of a twin or a stacking fault ribbon on the other side of the grain boundary [35,91,93,95]. As shown in Figure 14, the incident dislocations dissociate at the grain boundary, where some partials are absorbed and reflected. Other partials emit from the grain boundary into the adjacent grain and form a stacking fault ribbon. This phenomenon often appears in hcp metals which have a high tendency to form deformation twins. When the orientation of adjacent grain is appropriate to nucleate deformation twins rather than activation of slip systems due to the stress concentration induced by dislocation–grain boundary interactions, such phenomenon will occur.

#### 3.1.2. Dislocation Transmission Mechanisms across a Grain Boundary

The dislocation transmission across a grain boundary in coarse-grained metals is somehow still a controversial topic, and several criteria have been proposed to elucidate this phenomenon [20,29,34,35,37,40,93]. These criteria are found, confirmed or established based on different types of standpoints, such as the misorientation, the geometrical conditions, the resolved shear stress, the grain boundary dislocation energy and the SFE.

Among these criteria, it is found that the dislocation transmission should follow the common criteria of minimizing the Burgers vector magnitude of residual dislocation partials on the grain boundary and of maximizing the resolved shear stress magnitude of the slip systems [29,34,35]. These two conditions must be met simultaneously, since the resolved shear stress is the power to overcome the resistance of grain boundary while the transmission requires small residual dislocation Burgers vector existing in the grain boundary due to the smallest increase in strain energy of grain boundaries. Additionally, some other factors are also reviewed in details as follows.

Misorientation

The misorientation of a grain boundary has an obvious effect on the interaction behavior between dislocations and the grain boundary. It can serve as a rough criteria to predict the transmission resistance of grain boundary [16,19,96]. This may be ascribed to the fact that the deviation between different slip planes of the incoming and outgoing dislocations across high angle grain boundaries is larger than that for low angle ones.

A low angle grain boundary can actually impede dislocation s to some extent, but much weaker than a high angle grain boundary. The high angle grain boundary is a strong barrier, which impedes the slip of dislocations and makes them pile-up against the boundary. Since a residual dislocation should form in a grain boundary if a dislocation transmits it by the direct mode, additional energy is required and the dislocation transmission becomes difficult. On the other hand, a low angle grain boundary can be taken as a periodic array of dislocations. The interaction between dislocations and low angle grain boundaries is the interaction of dislocations with different Burgers vectors. When passing through a low angle grain boundary, jogs are left in the grain boundary while kinks are formed in the transmitted dislocations. This implies an increase in the total energy during the interaction process.

b.Geometrical condition

The grain boundary misorientation is a rough criteria to estimate the resistance of a grain boundary to dislocation motion. However, it cannot determine whether the dislocation can or cannot transmit across a grain boundary.

It has been found that the dislocation transmission strongly depends on the geometrical condition of a grain boundary, which includes the orientations of slip systems and grain boundary parameters. The earliest criteria of the geometrical condition is established by Livingston and Chalmers (*LC*) [97,98], which is defined as:*LC* = (*e*_in_ ∙ *e*_out_) ∗ (*g*_in_ ∙ *g*_out_) + (*e*_in_ ∙ *g*_out_) ∗ (*g*_in_ ∙ *e*_out_)(2)
where *e* and *g* are the slip plane normal and slip directions, respectively. The subscripts refer to the incoming and outgoing slip systems, respectively. It is supposed to predict the dislocation transmission if the value of LC is minimized. However, this criteria has been considered not sufficient to account for the operative slip systems.

Clark et al. [34,94,97] combined geometrical and resolved shear stress conditions together and proposed another criteria, which is expressed as:*M_c_* = (*l*_in_ ∙ *l*_out_) ∗ (*g*_in_ ∙ *g*_out_)(3)
here *l* is the line traces of two slip planes intersected on the grain boundary plane. The slip system with the maximum value of *M_c_* has the largest possibility to transmit across a grain boundary.

So far, several transmission factors have been widely used in order to predict the ease of dislocation transmission [20,29,35], such as *m′*, *LRB* (Lee–Robertson–Birnbaum) and the Schmid factor. The *m′* and *LRB* are developed from the *M* criteria and based on the observation of dislocation transmission by TEM [29,35], which can be expressed as:*m′* = cos *ψ* cos *κ*(4)
*LRB* = cos *θ* cos *κ*(5)
where *ψ* is the angle between the slip plane normal of the two crystals, *κ* is the angle between the Burgers vectors of incoming and outgoing dislocations and *θ* is the angle between the line traces of two slip planes intersected on the grain boundary plane [35]. Higher values of *m′* or *LRB* indicate larger probabilities of dislocation transmission, whose maximum value is 1 referring to the easiest transmission condition [20].

Among them, *m′* can be evaluated easily by an orientation map of electron backscatter diffraction (EBSD), but it is hard to evaluate *LRB* depending on a non-destructive experiment [35]. Additionally, it remains controversial which is better between *m′* and *LRB* [20]. The Schmid factor is based on the macro-scale stress state, which is not so appropriate for the dislocation transmission dominated by the local micro-scale stress state [29]. Thus, *m′* is considered as the most convenient one to predict the dislocation transmission.

To better predict the dislocation transmission by simulation, Bieler et al. [35] proposed the following modified models of *m′* and *LRB* combining the accumulated shear γ:(6)$$m_{\gamma }^{\prime }=\;\underset{\mathrm{in}}{\sum }\underset{\mathrm{out}}{\sum }m^{\prime }\left(\gamma_{\mathrm{in}}\gamma_{\mathrm{out}}\right)/\underset{\mathrm{in}}{\sum }\underset{\mathrm{out}}{\sum }\left(\gamma_{\mathrm{in}}\gamma_{\mathrm{out}}\right)$$
(7)$${LRB}_{\gamma }\;=\;\underset{\mathrm{in}}{\sum }\underset{\mathrm{out}}{\sum }LRB\left(\gamma_{\mathrm{in}}\gamma_{\mathrm{out}}\right)/\underset{\mathrm{in}}{\sum }\underset{\mathrm{out}}{\sum }\left(\gamma_{\mathrm{in}}\gamma_{\mathrm{out}}\right)$$

The sums go over all possible combinations of slip systems in the two grains along the grain boundary. Furthermore, the criteria combining the *m′* and *LRB* together can be represented as:(8)$$s_{\gamma }\;=\;\underset{\mathrm{in}}{\sum }\underset{\mathrm{out}}{\sum }\cos\psi \cos\theta \cos\kappa \left(\gamma_{\mathrm{in}}\gamma_{\mathrm{out}}\right)/\underset{\mathrm{in}}{\sum }\underset{\mathrm{out}}{\sum }\left(\gamma_{\mathrm{in}}\gamma_{\mathrm{out}}\right)$$

Similarly, the *m′* can be modified using the Schmid factor *m* on each slip system:(9)$$m_{m}^{\prime }=\;\underset{\mathrm{in}}{\sum }\underset{\mathrm{out}}{\sum }m^{\prime }\left(m_{\mathrm{in}}m_{\mathrm{out}}\right)/\underset{\mathrm{in}}{\sum }\underset{\mathrm{out}}{\sum }\left(m_{\mathrm{in}}m_{\mathrm{out}}\right)$$

c.Energy

Among all the criteria of dislocation transmission, energy is the dominant factor that determines whether a dislocation is able to transmit across grain boundary or not. This viewpoint is generally accepted although some aspects are still ambiguous and require further study.

In the case of leaving residual partials on a grain boundary during dislocation transmission, the accumulation of residual dislocations leads to the increase of grain boundary energy, which is expressed as the strain energy density [29,34,40,93]. This energy is determined by the magnitude of Burgers vector of the residual dislocations and should be minimized during dislocation transmission, as shown by Equation (1). The strain energy density dominates the probability and behavior of dislocation transmission. For instance, the dislocation can be piled up if it may generate a residual dislocation with a large Burgers vector, even though it satisfies the condition of maximum resolved shear stress. On the other hand, according to the atomistic simulation, a dislocation is also possible to transmit across a grain boundary easily if the dislocation interaction changes the local misorientation and reduces the grain boundary energy [96]. Moreover, it is found that the accumulative strain energy density with the increase of strain may activate more dislocation sources and change the Burgers vectors of emission dislocations [93].

The effect of grain boundary energy on the dislocation transmission resistance across a grain boundary is still controversial in the literatures. However, there are tendencies verified by experiments that the resistance of a boundary to dislocation motion is related to the complexity level and the interfacial energy of boundary structure. Dislocations may pile up at impenetrable boundaries with complex structure and high interfacial energy by absorption or dissociation without any transmission into the adjacent grain, such as high angle grain boundaries. Whereas transmission is easy to occur on some specific activated slip systems across the boundaries with simple and definite structure, such as low angle boundaries, low Σ boundaries and coherent twin boundaries [35].

The above understandings are the most-accepted concepts, however, an opposite conclusion has been reported by Sangid et al. [37]. They proposed a methodology to measure energy barriers for dislocation transmission and use it in simulation, which is found to be consistent with experimental observations of dislocation transmission and the *LRB* criterion. These results indicate that a grain boundary with a lower interfacial energy offers a stronger barrier against dislocation transmission. For instance, the Σ3 grain boundary (twin boundary) with low interfacial energy has a much larger resistance to dislocation motion than the other ones, while the Σ19 grain boundary has a high interfacial energy shows easy resistance of dislocation transmission.

#### 3.1.3. Other Influencing Factors

Dislocation type

Most dislocation transmission mechanisms are established regardless of the incident dislocation type; however, some studies show that this is something significant during the dislocation–grain boundary interaction. According to the detailed investigations of dislocation–twin boundary interactions, the type of incident dislocation dominate the transmission behavior. This evidence will be reviewed in the next chapter. Additionally, different responses are reported in the interactions of high angle grain boundaries with edge and screw dislocations in ultrafine-grained Al [43]. The incident screw dislocations are absorbed by high angle grain boundaries and exert weak internal stresses to the grain boundary. While the incident edge dislocations seem to be piled up in front of high angle grain boundary and forms high internal stress to the grain boundary. These observations are further confirmed by unloading experiments: the screw dislocations are still accommodated in the grain boundary instead of moving back to the sources during unloading, whereas the edge dislocations are able to move back to the sources indicating a different response to the grain boundary. These phenomena may imply that the research on dislocation–grain boundary interactions should not only focus on the characteristics of a grain boundary, but also concern about the incident dislocation types.

b.Strain and strain rate

Some recent research shows that the practical dislocation transmission across a grain boundary during deformation could be more complicated than what is known. It was found that the successive dislocation transmissions may destroy or interrupt the original structure of a grain boundary and form local misorientations, which results in activating more dislocation sources and changing the interaction behavior at the intersection point [29]. For example, the dislocation emission from a grain boundary may change from partial dislocations to perfect ones with increasing strain [93]. This change during dislocation transmission is always not considered in most studies but important for the transmission mechanism.

On the other hand, it is interesting that, under some situations, the individual high angle grain boundaries show weak influence on the stress–strain curve of pillar samples, whose curves are similar to the ones of single crystal samples whenever the slip transfer occurs or not [17,20]. It seems that these results are just the opposite to the common understanding that a high grain boundary is a strong barrier to dislocation motion. However, there are two main reasons to elucidate this conflictive phenomenon. The macroscopic mechanical properties are the average performances of both matrix and grain boundaries. It is hard to obtain a remarkable difference in macroscopic performances between samples of single crystal and bi-crystal that have only a single grain boundary. Moreover, it can be explained by the strain rate sensitivity (SRS) of the grain boundary [15,17]. The strain rate dependence of copper pillars containing a penetrable high angle grain boundary via in-situ compression tests at different strain rate are analyzed [17] and it is found that the grain boundary act as different roles in dislocation transmission when the strain rate changes. It is easy for a dislocation to transmit across a grain boundary at low strain rates, whereas pile-ups and dislocation multiplication always occur in front of a grain boundary at high strain rates. When low strain rates are applied, a similar yield stress is obtained in both single crystal and bi-crystal samples. However, the bi-crystal sample has higher yield stress at high strain rates owing to the difficulty of dislocation transmission. Therefore, due to no grain boundary interaction, the SRS of the single crystal sample is almost constant, which is distinguished with the SRS of bi-crystal samples. The difference of SRS between single crystal and bi-crystal samples is ascribed to the reorientation of dislocations at the grain boundary during transmission, which is hard to occur at high strain rate. Thus, a high SRS is found in bi-crystal samples as a result of dislocation–grain boundary interaction, while single crystal samples have a relatively low SRS due to dislocation–dislocation interactions and formation of dislocation networks.

This research indicates there are a large number of factors influencing the process of dislocation–grain boundary interactions in coarse-grained metals, which includes energy during interaction [34,35,37,93], the type and orientation (geometrical factors) of boundaries and dislocations [16,19,20,35,43] and strain and strain rate [15,17,93]. These factors may affect each other and their correlations are complicated. Since most researches focused only on one factor regardless of other important factors, this may lead to one-sided and incomplete conclusions. For example, the grain boundary interfacial energy has been considered as one of the dominating grain boundary characteristics influencing the dislocation–grain boundary interaction. However, this factor cannot be studied independently from the dislocation types, because even the same grain boundary may exhibit a varied response to different types of incident dislocations. Any isolated study will draw contradictory conclusions with the others, as described in Section 3.1.2.

### 3.2. Dislocation–Grain Boundary Interactions in Ultrafine-Grained and Nano-Grained Metals

Most researches of dislocation–grain boundary interactions narrated in the above sections are conducted in conventional metals with large grains above several micrometers. In these coarse-grained metals, the dislocation–grain boundary interaction plays a significant role in their mechanical properties. Owing to the impediment of grain boundary to dislocation motion, the plastic deformation of metals with relatively fine grain size requires larger applied stresses, which is known as the Hall–Petch relation [99,100]. In fact, the required stress for dislocation transmission across a grain boundary is much higher than the yield stress of a metal, which can reach four times the value of yield stress [50]. This is because the yield stress can be considered as an average of the contributions from grain boundaries and interior matrix.

When the grain size is smaller than 1 μm, i.e., ultrafine-grained and nano-grained metals, whose grain sizes are considered as 250–1000 nm and 1–250 nm, respectively [41]. The deformation mechanisms and dislocation–grain boundary interactions of metals with grain sizes at these length scales are different from those of conventional coarse-grained metals.

For ultrafine-grained and nano-grained metal, grain boundary mediated mechanisms, such as grain boundary sliding and rotation, have a large influence on the deformation mechanisms [41,42,43]. The limited distance between the adjacent grain boundaries generates a strong image force field which suppresses dislocation motions. The large resistance results in a confined number of dislocation pile-ups and requires larger stress to increase this number. With the decrease of the grain size to nano-scale, the number of pile-up eventually reduces to one [41]. Since the Hall–Petch relation is established according to dislocation pile-ups in front of grain boundaries, it does not work in some nano-grained metals and even shows an “Inverse Hall–Petch” phenomenon due to the activation of grain boundary sliding and rotation.

A model is proposed to interpret the dislocation–grain boundary interaction in nano-grained metals, which is called “core and mantle” [41]. The core refers to the grain interior with a relatively homogeneous stress state and area free of dislocations, while the mantle is the grain boundary region. Initially, the dislocations are likely to nucleate at grain boundaries and move inside the mantle region rather than emit into the grain interior, as shown in Figure 15a. This plastic deformation of the mantle region is much more severe than the core and leads to a fast increase of dislocation density. Due to the breakdown of dislocation pile-ups, the hardening contributor of the core is much lower compared with that of the mantle region, where dislocations are accumulated and cross-slip systems are active ultimately.

At high strains, dislocations from grain boundaries is able to emit into the grain interior and glide forwards until absorbed at the opposite grain boundary, as shown in Figure 15b. At the same time, the grain boundary sliding and rotation are able to activate. The emitted dislocations in ultrafine-grained and nano-grained metals always show a curvature configuration than those in coarse-grained metals. This is due to the strong image force derived from grain boundaries close to each other. The process of dislocation absorption is accompanied by an increased number of grain-boundary dislocations, rearrangement of grain-boundary dislocations, and a change in the grain-boundary structure. Further strain increases will activate more dislocation sources and the emission process is continuous. Additionally, the dislocation transmission is also observed in ultrafine grains [43]. These indicate that the dislocation–grain boundary interactions in ultrafine-grained and nano-grained metal have some similar behaviors with those of coarse-grained metals when the incident dislocation impinges on grain boundaries.

### 3.3. Dislocation–Twin Boundary Interactions in Coarse-Grained Metals

#### 3.3.1. Basic

There are two types of twin boundaries: coherent and incoherent ones. For the incoherent twin boundaries in coarse-grained metals, their interactions with dislocations show no different behaviors compared with grain boundaries [12,18]. For instance, the dislocations are able to transmit across incoherent twin boundaries due to applied stresses, and leave residual dislocations or form steps on the boundaries. This indicates that the incoherent twin boundary is not an impenetrable boundary to dislocation motion. Additionally, the incoherent twin boundary does not have a remarkable effect on the mechanical properties of metals, since the volume fraction of the coherent twin boundary is much larger than that of incoherent one in most cases. For example, the spacing of incoherent twin boundaries in nanotwinned Ag is at least an order of magnitude larger than the spacing of coherent twin boundaries, which shows a weak influence on the mechanical properties. Thus, most of the researches focus on the coherent twin boundaries. The following contents are engaged in these researches and the term “twin boundary” refers to the coherent twin boundary.

Since the coherent twin boundary is an Σ = 3 grain boundary, the dislocation–twin boundary interaction shows some similarities with the dislocation–grain boundary interaction. However, compared with the complicated and varied grain boundaries, more intensive and detailed studies can be done regardless of other influencing factors derived from twin boundaries and only focus on factors such as dislocation types and the relationship between the slip planes and the twin plane [27,29], owing to its definite mirror symmetry structure. When the dislocation approaches towards the coherent twin boundary, it tends to be parallel to the cross-line between the slip plane and the twin boundary before impingement. Then, it may be absorbed by the twin boundary through accommodation, reconstruction or dissociation into glissile and sessile partials. In some proper conditions, direct and indirect dislocation transmissions across twin boundaries are able to occur due to the increased applied stresses.

Figure 16 shows an example of dislocation pile-ups before transmission across a coherent twin boundary. It indicates that dislocations are primarily impeded by the coherent twin boundary until the stress reaches a critical value [101], below which dislocations are absorbed in the coherent twin boundary, and above this value the dislocation can transmit into the twinned region. In fcc metals, the critical stresses are estimated to be different for screw and non-screw dislocation, which are about 400 MPa [101,102,103] and 1 GPa [104,105], respectively. The critical stress value is found to be influenced by factors such as the SFE, the shear modulus and the Shockley partial Burgers vector.

When a screw dislocation glides parallel to the coherent twin boundary, direct transmission of dislocation may occur by means of cross-slip without leaving any residual dislocation on the twin boundary, as shown in Figure 17. In this case, the coherent twin boundary is the common intersection line of the slip systems on either side of the twin boundary [40]. Although no dislocation pile-ups form in front of the coherent twin boundary, the dislocations cannot transmit across directly, but are absorbed firstly by the coherent twin boundary and then emit into the neighbor twin interior.

In other cases, dislocations may dissociate into glissile and sessile partials when impinging on the coherent twin boundary. The sessile partials may form steps on the twin boundary while the glissile partials shall glide in the twin boundary leading to the migration of the twin boundary and resulting in thinning or thickening of the twin [27,29]. This sort of direct dislocation transmission may generate residual dislocations to form steps on the coherent twin boundaries, which is a cyclic and continuous process with the increase of strain [12,106]. For example, Figure 18 shows the TEM images of three dislocations transmitting across a coherent twin boundary and leaving a sharp step with three layers of atoms. The transmission process proceeds as the following sequences. The incoming dislocation glides to the coherent twin boundary and dissociates as: full dislocation → Frank dislocation + glissile Shockley partial. Then, the glissile Shockley partial glides away along the twin boundary, while the Frank dislocation further dissociates as: Frank dislocation → full dislocation + sessile Shockley partial. Finally, the full dislocation emits to the other side of the coherent twin boundary leaving a step of sessile Shockley partial.

In TiAl alloy [39], when an incoming full dislocation with a Burgers vector of 1/2 <011> transmits across the coherent twin boundary, a residual sharp step dislocation is in the coherent twin boundary. This step dislocation may have the same Burgers vector as the full dislocation (1/2<011>). In some other cases of this study, the incoming full dislocation can either dissociate to a glissile 1/6[112] twinning dislocation and a sessile 1/3[$1\bar{11}$] Frank dislocation, or dissociate to a glissile 1/6[112] twinning dislocation and a sessile 1/6[211] Shockley dislocation. These phenomena indicate that the behavior and products of dislocation–twin boundary interaction in the coarse-grained metals may be different from those of the dislocation–grain boundary interaction due to some influencing factors, which are going to be discussed in the following sections.

#### 3.3.2. Dislocation Transmission Mechanisms across a Coherent Twin Boundary

Since the coherent twin boundary structure in one metal is the same, the different responses to impinged dislocations are almost due to different types of incident dislocations and different relationships between the slip planes and twin planes. Furthermore, the SFE plays a significant role in the dislocation–twin boundary interaction.

Dislocation type

The dislocation–twin boundary interaction strongly depends on the types of incident dislocations [107]. The discrepancy of twin boundary resistance against dislocation motion may be extremely large [105]. For example, in fcc metals, the coherent twin boundary is a strong barrier to the non-screw dislocations, however, screw dislocation can transmit across a coherent twin boundary directly by cross-slip without leaving residual steps on the twin boundary, analogue to the Friedel–Escaig mechanism for cross-slip [40,108]. When non-screw dislocation impinges on a coherent twin boundary, it is more favorable to dissociate into partial dislocations on the interfacial plane rather than transmit to the other side of the twin boundary. A large number of non-screw dislocation pile-ups are found at the coherent twin boundary after deformation, and the transmission of non-screw dislocation requires high stress and a large accumulation number of dislocations [31]. The critical stress required to transmit across a coherent twin boundary can be as high as 1 GPa when the incident dislocation has a non-screw characteristics [104,105], while the pile-ups of screw dislocations are scarce which exert weak back stresses on the active dislocation sources [31]. The critical stress for screw dislocation transmitting across coherent twin boundary is estimated to be between 300 and 400 MPa [101,102,103].

In recent studies [31,105], the dislocation transmission mechanism in fcc metals is classified specifically into three categories according to the types of dislocation impinging on the coherent twin boundary: the cross-slip mode, the hard mode and the soft mode.
The cross-slip mode denotes the transmission of perfect screw dislocations whose Burgers vector is parallel to the coherent twin boundary plane. In this mode, the screw dislocations are usually derived from grain boundaries, which are always activated favorably by applying tension/compression parallel to the coherent twin boundary. Then, they may transmit across a twin boundary by cross-slip without leaving residual dislocations and are ultimately impeded by the grain boundary at the opposite side;For the hard mode, the Burgers vector of incoming dislocations is inclined to the coherent twin boundary plane. This may require residual dislocations left in the twin boundary and a higher critical stress to transmit across a coherent twin boundary;Dislocations with a slip plane parallel to the twin boundary are defined as the soft mode, which occurs when the maximum shear stress is parallel to the coherent twin plane. The dislocations may glide on the coherent twin boundary leading to migration of the twin boundary by twinning or detwinning.

b.Geometrical condition

Additionally to the dislocation type, the relationship between the slip plane and the twin plane also affects the behavior and products of the interaction [39,109]. The dislocation–twin boundary interaction can be different due to different slip systems or deviation between slip planes and twin planes even the dislocation type is the same. For example, two variants of $\left\{111\right\}\frac{a}{2}<011>$ of slip systems in 304 stainless steel, which are (111)$\frac{a}{2}\left[110\right]$ and (111)$\frac{a}{2}\left[101\right]$, are reported to show different responses when impinging on the coherent twin boundary, respectively [29,93]. The (111)$\frac{a}{2}\left[110\right]$ dislocations can transmit across the twin boundary while the (111)$\frac{a}{2}\left[101\right]$ dislocations just glide along the twin boundary rather than transmit it.

It is supposed that the dislocation types and geometrical conditions have a combined effect on the dislocation–twin boundary interaction [29]. In fcc metals, Zhu et al. [27] summarized four possible types of dislocation–twin boundary interactions: the 30° Shockley partial, the 90° Shockley partial, the screw perfect dislocation, and the 60° perfect dislocation. The screw perfect dislocation can transmit across the twin boundary by the cross-slip mode. The other types of interactions can either transmit across twin boundaries by the cross-slip mode or hard mode under different conditions. The alternative behaviors are considered to be determined by the SFE of metals, which is summarized in the next chapter.

c.Stacking fault energy

The effects of SFE on the dislocation–twin boundary interaction are considered in two aspects [31]. The decrease of SFE may hinder the dislocation transmission across a coherent twin boundary by the cross-slip mode and increase the resistance of dislocation transmission by the hard mode. This is because the critical stresses required for nucleation, dissociation and constriction of perfect and partial dislocations are strongly related to the SFE, which further affects the dislocation interaction during transmission. High SFE encourages the constriction of Shockley partials into the screw type and facilities the cross-slip mode transmission. Whereas the low SFE increases the critical stress for dislocation emissions by the hard mode transmission, which leads to the increase of twin boundary resistance against non-screw dislocations. The detailed explanations can be found in the study by Liebig et al. [31].

The above assumptions are supported by the experiments on Cu and CuZn30 alloys. The Cu sample with a relatively high SFE activates more slip planes due to the favorable cross-slip mode dislocation transmission across the coherent twin boundary during compression than those in the CuZn_30_ sample. Besides, the slip transfer steps across the coherent twin boundary show a smooth morphology in the Cu sample. While in the CuZn_30_ sample, the slip transfer exhibits discrete and well-defined features and the serrated stress-strain response.

### 3.4. Dislocation–Twin Boundary Interactions in Nanotwinned Metals

Since the twin boundary is a barrier to dislocation motion, the nanotwinned boundaries confine the slip of non-screw dislocations perpendicular to the twin plane due to the limited spacing between two adjacent twin boundaries. Thus, the dislocation–twin boundary interaction in nanotwinned metals is different from that in coarse-twinned metals. Besides, the dislocation–boundary interactions in nanotwinned metals are also diverse from that in nano-grained ones. This is ascribed to the spacing between grain boundary is typically on the micro-scale in nanotwinned metals, while the spacing between twin layers is much smaller, which is on the nano-scale [105]. In this case, the twin spacing governs the strengthening of metals whereas the micro-scale grain boundary contributes much less strength, i.e., the bulk strength increases with the decrease of twin spacing [18].

Owing to the strong suppression from nanotwinned layers, dislocations are found to accumulate and glide in the vicinity of twin boundaries, which is likely the favored place for plastic deformation [44]. More specifically, the dislocations are favored to glide on the coherent twin boundaries by the soft mode, which results in twinning or detwinning due to migration of twin boundaries [105]. At the same time, dislocations tend to glide and accumulate in directions parallel to the twin boundaries rather than perpendicular to them. The dislocations glide towards twin boundaries may form Lomer–Cottrell (L-C) locks and resist the transmission of dislocations across a twin boundary [110]. Additionally, different from the large-spaced twin boundaries at the micro-scales, the progressive transmission of dislocations across the nano-scale coherent twin boundary by the hard mode may result in the severe distortion and loss of coherency on the twin boundary due to the formation of steps by residual sessile dislocations [45,111]. These changes of coherent twin boundaries lead to an increase of resistance to dislocation penetration and ultimately evolve into a distorted grain boundary with a high energy [44,46].

A recent study of nanotwinned Ag has found a possible reason responsible for the high strength of nanotwinned metals [18]. In fact, in some studies investigated by in-situ SEM tests [15,31,51,112], there is a ubiquitous phenomenon remaining unresolved: The strengthening contributed from dislocation transmission of individual twin boundary by the cross-slip mode is unexpectedly small, which is not consistent with the high strength obtained in nanotwinned metals [113]. Kini et al. proposes a possible explanation on this topic by testing nanotwinned Ag specimens with different twin spacing from 18 nm to 1800 nm. It is found that the strength increases dramatically when the twin spacing is smaller than 100 nm. This dramatic strength increase is found to come from the local curvature of the partial dislocations between two adjacent coherent twin boundaries, as shown in Figure 19. Since the cross-slip mode transmission requires the constrictions of partial dislocations into the perfect screw dislocations when transmitting across the coherent twin boundaries, the reduction of twin spacing leads to the increase of local curvature of partial dislocations, which further increase the resistance of dislocation transmission by the cross-slip mode. This is different from the observations in the twinned metals with a micro-scale spacing: the strengthening is contributed from the dislocation absorptions and the emissions during transmission across the coherent twin boundaries.

### 3.5. Dislocation–Phase Interface Interactions

The phase interface is supposed to have a much stronger capability on impeding dislocation motion than those of grain boundaries and twin boundaries due to the fact that dislocations must not only overcome the image force and accommodate the complex interface structure, but also accommodate different Bravais lattice, orientation and lattice parameters of the other phase. This stronger impediment makes the interaction between dislocations and phase interfaces different from the other kinds of interactions mentioned above.

Among all the influencing factors, the atomic bonding is likely to be the strongest one. The interface of non-metallic compound phases is a much stronger obstacle against dislocation motion compared with the interface between two different metallic phases, which is due to the complex incoherent interfaces with the poor matching of atoms and the high-indexed planes. Additionally, the interactions between dislocations and phase interfaces may be affected by the phase morphologies and dimensions, as well as the alignment or deviation of slip systems in different phases. In the following sections, the dislocation–phase interface interactions of two morphologies: particle and lamella, are reviewed.

#### 3.5.1. Particle

Particle at the nano-scale

The dislocation interaction with particle interface is governed by the particle size. There is a critical size determining this interaction, which is about 2–7 nm for non-metallic compounds [21]. When the size of the particle is smaller than the critical value, the dislocations are able to penetrate across the phase interface and glide through the whole crystal, which is called shearing. If the size is larger than the critical value, the dislocations may pass around the particle and form dislocation loops, which is termed as looping, as shown in Figure 20. No matter looping or shearing, both interactions generate new defects by consuming the deformation energy and increase the strength of matrix.

The process of shearing has three steps with the increase of applied stress [11]. Initially, the dislocations pile up in front of an interface, and the follow-up dislocations make the ahead dislocation curved along the interface. Then, due to the stress concentration, several dislocations reconstruct to be dislocations or superdislocations accommodating to the Bravais lattice, orientation and lattice parameters of the particle, and eventually glide through the whole particle. The shearing may lead to a large displacement or superlattice intrinsic stacking fault in the particle. As shown in Figure 21a–c, when the particle is partially sheared by dislocations, some regions become blurred due to the local displacements induced by dislocation shearing, compared with the crystal lattice before deformation [21,114]. When the particles are sheared by the arrays of dislocations, the displacements in the particles can be easily observed, as the steps shown in Figure 21d. Compared with the dislocation mobility in the metallic matrix, the mobility in the non-metallic particles is low due to the strong covalent bonds and high melting point of the particles [115,116].

b.Particle at the micro-scale

For particles with the sizes above 1 μm, the dislocation is more likely to be impeded in front of the interface rather than penetrating it [48]. The dislocations can only transmit the phase interface between two different metallic phases, since few dislocations are able to penetrate the phase interface of non-metallic compound particles with a size larger than 10 nm. At this dimension, the slip transfer across the phase interface between two different metallic phases is dominated by the dislocation transmission instead of shearing.

Since the critical resolved shear stress to transmit across a phase interface is much higher than grain boundary, the stress state plays a more important role in dislocations-phase interface interactions: the Schmid factor seems to be one of the possible reasons for the dislocation transmission but not sufficient to predict it [48]. At the same time, the alignment or deviation of slip systems of different phases appears to be another important criteria for the dislocation transmission.

#### 3.5.2. Lamella

The dislocation–phase interface interaction of a lamella depends on different atomic bondings on both sides of an interface. The dislocation–phase interface interaction between two metallic phases is similar for both particles and lamellae. Their dislocation transmission is strongly influenced by the alignment or deviation of slip systems of different phases. Whereas the slip transfer across the non-metallic interface always leads to the yielding of non-metallic phases by fracture or deformation.

Interface between two different metallic phases

The resistance of lamellar interface between two different metallic phases can be diverse due to the alignment or deviation of slip systems between two metallic phases and the annihilation speed of the residual interfacial dislocations produced by dislocation transmission. Dislocations are easy to transmit across the interface when the two phases have a small misorientation between slip systems. For example, the dislocation transmission across the α–β lamellar interface of Ti alloy shows distinct morphologies when the incident dislocations belong to different slip systems [47,49,118]. As shown in Figure 22a, the β lamellae (white laths) can be sheared by the prism slip. However, no shearing of β laths occurs for the basal slip (Figure 22b). This phenomenon was further investigated by Savage et al. [47] through the TEM observations of dislocations in front of the α–β lamellar interface. They found that few pile-ups of prism dislocations were observed in front of the α–β interface. In contrast, the basal dislocations with large distortions are accumulated in front of the α–β interface. The reason is due to the prism slip in the α matrix have a smaller deviation with the slip in the β phase compared with the situation for basal slip. Additionally, the fast annihilation of residual interfacial dislocations during the prism dislocation transmission was considered to further facilitate this process. Thus, the transmission resistance to basal slip is relatively stronger than that of prism slip.

b.Non-metallic compound lamellar interface

The dislocation interactions with lamellar interfaces between metals and non-metallic compounds are complicated. Since the dislocation–phase interface interaction of high-carbon pearlite steel has been comprehensively studied, it is taken as an example here.

In a macroscopic view, the deformation process of pearlite occurs as following:Elastic deformation occurs in both ferrite and cementite phases;Plastic deformation occurs in the ferrite lamellae in advance from the dislocation sources at the ferrite–cementite interfaces [119,120,121] and then proceeds into the cementite lamellae [122,123];The dislocations slip in each lamella interior independently and then the slip transfers appear by the yielding of cementite lamellae.

This process proceeds with a reorientation of pearlite colonies and a reduction of lamellar spacing [124,125,126,127]. It is supposed that the ferrite–cementite interfaces serve as the dislocation sources and strong barriers against dislocation motions, which is confirmed by TEM observations [128,129,130]. Thus, at the first stage of plastic deformation, it is assumed that dislocations in ferrite lamellae nucleate firstly at the interface, and then dislocations in cementite lamellae may nucleate at the interface.

In contrast with other dislocation–boundary interactions, the resistance of a phase interface against dislocation motion is much stronger. The impediment of the interface is strong enough to hinder the dislocations moving perpendicular towards the lamellae, which make them piled up or pinned at the interface, so that dislocations are constrained to move parallel to the lamellae. In this case, the dislocation may rotate its moving direction parallel to the interface leaving “superkinks” at the interface and continue to propagate without any increase in the total dislocation length [28]. Therefore, the dislocations glide independently in their own lamellar interiors before slip transfer.

The slip transfer of a ferrite–cementite interface is also distinguished from those of other interfaces mentioned above [10]. The slip transfer occurs always from ferrite to cementite rather than from cementite to ferrite [122,131], which is induced by stress concentration at sites of dislocation pile-ups. The slip transfer occurs along slip systems of 1/2{110}<111> and 1/2{112}<111> in ferrite [10,30,132,133]. However, for cementite, the slip transfer leads to the yielding of cementite lamellae by fracture or bending by slip steps, as shown in Figure 23. For coarse cementite lamellae, the slip transfer is always by a fracture due to its brittleness [22], as shown in Figure 24. While both fracture and bending by slip steps may occur for fine cementite lamellae due to its deformability.

The mechanism of dislocation transmission across the ferrite–cementite interface is still a dubious topic. Several models have been proposed and simulations have been done to predict this probability, but no persuasive conclusion has been drawn. There is few solid evidence of the dislocation transmission across a ferrite–cementite interface by means of direct transmission, although the slip transfer has been observed in the pearlite. The feasibility of dislocation transmission is researched by using *LRB* criteria [132,133], dislocation transmission mechanism *m′* (mentioned in Section 3.1) [30] and simulation [122,131,134]. According to *LRB* criteria, two {110}<111> slip systems and one {112}<111> slip system of ferrite are possible to transmit across the ferrite–cementite interface. The dislocation transmission mechanism *m′* of grain boundary can be also applied.

Furthermore, most scholars consider that the ferrite–cementite interface is a strong barrier of dislocation motion, especially for dislocations with the slip direction perpendicular to the interface. At the early stage of deformation, dislocations in ferrite lamellae pile up in front of the interface and slip transfer occurs eventually owing to the increase of stress concentration. However, some researchers argue that dislocation transmission across the ferrite–cementite interface is relatively easy at the early stage of deformation. The subsequent dislocation pile-ups are due to the changes of cementite structure induced by dislocation transmission. Zhao et al. [123] claims that dislocations can easily transmit from a ferrite lamella to a single crystalline cementite lamella across a flat ferrite–cementite interface. Then, the single crystalline cementite lamellae subdivide into polycrystals. The interface of polycrystalline cementite lamellae can effectively impede the dislocation motion. Therefore, the dislocation transmission mechanisms across the ferrite–cementite interface are proposed according to calculations and simulations, which are diverse and ambivalent requiring further studies and experimental verifications by advanced characterization methods such as high-resolution and in-situ TEM mechanical testing [135].

Dislocations-phase interface interactions play an important role for structural metals, in not only the strength [136,137], but also other mechanical properties such as ductility and toughness [138], wear resistance [139], fatigue properties [140,141,142,143] and micro-pitting [144], as well as the properties of heterogeneous structures [145,146,147]. All these need the systematic investigations with exquisitely designed experiments and/or simulations.

## 4. Interactions between Deformation Twin and Boundary

Since the twin nucleation and growth are driven by the motion of twin boundary dislocations, the deformation twin-boundary interaction is considered as the result of interaction between the incident twin boundary dislocations and the obstacle boundary [13]. Thus, it shares many similar characteristics with the dislocation–boundary interactions. However, the transmission of deformation twin across a boundary may be much harder than a single dislocation. This is ascribed to several possible reasons:Since the twin nucleation and growth are the result of partial dislocation motion, the dislocation slip must be active prior to the twin formations and the stress level for the twin-boundary interaction is higher compared with that for the dislocation–boundary interaction [148];Since the emitted twin keeps the same variant as the incident one, the resolved shear stress and slip system of the adjacent grains should be favorable for the twin nucleation of the same variant. In this case, both the crystals and twin pairs on each side of the boundary should have small misorientation.

On the other hand, for the twin–twin interaction, since the twin boundaries are fast channels for dislocation motions, the applied stress can be released by dislocation gliding on the twin boundaries from the deformation twin to the obstacle twin instead of forming twin transmission. Thus, the twin transmission can only occur under limited conditions.

### 4.1. Deformation Twin-Grain Boundary Interactions in Coarse-Grained Metals

#### 4.1.1. Basic

Since hcp metals are favorable for the formation of deformation twins, they have been taken as one of the model metals to study the deformation twin-grain boundary interaction, especially pure Mg and Mg alloy. Similar results have been obtained in these studies [14,32,36,149,150,151] and several factors are engaged in the behavior of twin transmission across a grain boundary in coarse-grained metals, which has many characteristics in common with dislocation–grain boundary interaction.

The process of twin transmission in coarse-grained metals proceeds as the following sequences. The deformation twin propagation is primarily blocked by the grain boundary, which induces stress concentration in the vicinity. With the increase of stress concentration, the stress of the deformation twin relaxes in the neighboring grains and a new twin nucleates on the other side of the grain boundary due to local strain compatibility [151], as shown in Figure 25.

#### 4.1.2. Twin Transmission Mechanisms across a Grain Boundary

Misorientation

The misorientation has been considered as the primary factor determining the twin transmission in coarse-grained metals [14,32,36]: Deformation twins can transmit low angle grain boundaries but unlikely to do so when impinging on a high angle grain boundary. The impediment of twin propagation may cause high back stress and stress concentration in the vicinity, which generates local non-basal slip and secondary twinning. Twin transmission always occurs in the case of crossing a low angle grain boundary. The speed of twin transmission across low angle grain boundary is rapid, which is observed by in-situ SEM and in-situ optical microscopy (OM) mechanical testing [149,150].

Twin transmission determined by misorientation is probably due to the favorable twin nucleation conditions. Compared with high angle boundaries, low angle grain boundaries are composed of dislocation arrays, which are assumed to be favorable for activation of twin boundary dislocation and nucleation of deformation twins [14]. Furthermore, small misorientation facilitates twin nucleation due to a similar slip system and a sufficient resolved shear stress. With the increase of misorientation, the tendency of twin nucleation in the adjacent grain decreases due to the maximum resolved shear stress deviating away from the favorable slip systems [36].

b.Other influencing factors

Besides misorientation, the geometrical condition and chemical composition may also affect the behavior of twin transmission across grain boundaries in coarse-grained metals.

In most transmitted twin pairs, the angle between their twin plane normals tends to be minimized and the twin variant of incoming and outgoing twins should be maintained constant across the grain boundary [36]. These mechanisms are consistent with the criteria of misorientation. For low angle grain boundaries, the angle between twin plane normals of the incoming and outgoing twins is the misorientation of the grain boundary. The tendency of twin transmission increases with the reduction of misorientation owing to the increase of resolved shear stress. In this case, the easy slip system activated by the resolved shear stress must be the same as the incident twin. Thus, both the incident twin and emitting twin have the same variant.

The critical misorientation angles of twin transmission are slightly different for different metals. For instance, the highest transmission frequency in Mg is found to be the misorientation angle with a value below 15°–20° [32]. For pure Re, twin transmission is much easier to happen compared with Mg, leading to a relatively high critical misorientation angle [14,36]. This is ascribed to the low twin boundary energy and different deformation twinning system of Re.

### 4.2. Deformation Twin-Grain Boundary Interactions in Nano-Grained Metals

The deformation twin-grain boundary interaction in nano-grained metals shows distinct behaviors compared with that in coarse-grained metals. It is hard for a deformation twin to grow and intersect with grain boundary in nano-grained metals, let alone its transmission across a grain boundary [46]. Once the deformation twin nucleates at the grain boundary, the twin can grow into the grain interior due to the applied stress but is limited to a certain size. This is because the twin tip is imposed by a high image stress derived from grain boundaries nearby in nano-grained metals, which hinder the propagation of the deformation twin. At the same time, the increasing applied stress leads to nucleation of dislocations both from grain boundaries and twin boundaries. The progressive dislocation–twin boundary interactions result in severe distortion and loss of coherency on the twin boundary, which may eventually evolve into a distorted grain boundary with a high energy.

### 4.3. Twin–Twin Interactions

Although the coherent twin boundary has a definite structure, the twin–twin interaction is complicated and shows unique features compared with the deformation twin-grain boundary interaction. For example, in contrast to the dislocation gliding in the matrix, the twin boundaries can serve as the fast channels for dislocation motions due to no image force derived from the obstacle boundary [152]. Once the twin intersection is established, it is easy for dislocations to move from the twin boundaries of the incident twin to the twin boundaries of the obstacle twin. In this case, the applied stress can be released by this way rather than forming a twin transmission.

Furthermore, the twin–twin interaction behavior depends on the Bravais lattices of metals: The twin transmission is available to occur in bcc and fcc metals under suitable conditions [39,148], while the twin transmission is considered to be impossible for hcp metals, such as Mg [153,154]. These differences are reviewed in the following sections, separately.

#### 4.3.1. Twin–Twin Interactions in bcc and fcc Metals

There are two types of twin–twin interactions in bcc and fcc metals [39,148]. Figure 26 shows an ordinary twin transmission in bcc and fcc structural metals. The process proceeds as follows:The incident twin impinges on one side of the obstacle twin boundary and the incident twin propagation along the original direction is blocked;The incident twin transmits the obstacle twin boundary and alters the propagate direction along the close-packed plane of the obstacle twin until it approaches the other side of twin boundary;The incident twin transmits the twin boundary again and continue to propagate along the original direction beyond the obstacle twin.

The other type of intersection may generate a local distortion as well as a low angle grain boundary at the position of the twin–twin intersection, and the twin boundaries disappear at the intersection position [39]. Additionally, the crystalline lattice of the twin and the matrix distort to some degree and a low angle grain boundary is formed due to the complex reaction.

#### 4.3.2. Twin–Twin Interactions in hcp Metals

Structure of the twin–twin intersection

The twin–twin intersection in the hcp metals can result in the large lattice deviation in the vicinity of twin–twin intersection and forms a unique structure, which consists of an incoherent interface and a step-like faceted structure [13,38]. The TEM images of a twin–twin intersection is shown in Figure 27. The intersection forms a complex step-like faceted structure, which is composed of an incoherent interface (twin–twin boundary marked as B-B or P-P) and several facets connected with the twin–twin boundary, marked as B-P or P-B. The B and P represent the basal and prismatic planes, respectively. The twin–twin boundary is often aligned as the basal plane of one twin nearly parallel to the basal plane of another twin with a small misorientation across the boundary (B-B boundary), or similarly, the prismatic plane of one twin is nearly parallel to the prismatic plane of another twin with a small misorientation across the boundary (P-P boundary). The structure of facets has the characteristics that the basal planes in the matrix almost align with the prismatic planes in the twin (B-P facet), or similarly, the basal planes in the twin align with the prismatic planes in the matrix (P-B facet). There are many facets in the vicinity of the twin–twin intersection due to the large lattice deviation, while the number of facets drops dramatically at the remote area from the twin boundary.

It is found these facets play a key role in the twin boundary migration, because the facets must move with the twin boundary propagation and interact with twin boundary dislocations [13]. They may facilitate the twin boundary dislocation motion and potential atom shuffling across the twin boundary, which improves the twin boundary mobility.

The reason for the formation of the faceted structure is that the actual twin plane of Mg metal has a small deviation from the theoretical ideal twinning system ({1012}<1011>) [38]. Compared with geometrically and energetically unfavorable twinning planes, the interfacial energy to accommodate this deviation by formation of small-scale facets along twin boundaries is lower [13].

b.Behaviors during the twin–twin interaction

During the process of an incident twin impinging on an obstacle twin, a large number of dislocations are engaged [13]. Initially, the tip of the incident twin exhibits a sharp morphology when it is far away from the intersection point. Then, when the twin tip approaches the obstacle twin, it becomes blunted due to a high stress in front of the tip. Finally, a step forms at the intersection place of the obstacle twin boundary once the incident twin impinges on the obstacle twin.

After the twin–twin intersection, twins may grow thicker and coalesce together with the increase of applied stress [155]. As shown in Figure 28, the intersected twins grow thicker and the twins with the same variant may coalesce together once they meet with each other at low strains. At high strains, the twin variants with the highest Schmid factors may grow faster and larger than the others and take over the other twin variants eventually. The twin thickening is due to the twin boundary dislocation motion and twin boundary migration induced by the applied stress [153], as shown in Figure 29. The twin boundary dislocation glides from one twin boundary towards the other twin boundary with the stress increase. These dislocations may dissociate at the twin–twin intersection and glide on the new twin boundary leaving a tilt boundary behind, which results in the growth of the twin and the extension of the tilt boundary.

c.Interaction mechanisms

Notably, the twin transmission is unlikely to happen in the hcp metals in most cases [153,154]. It may be ascribed to two reasons. Since the twin formation requires the activation of proper twin boundary dislocations, the limited number of slip systems in hcp metals also means the limited number of twinning systems. In this case, the resolved shear stress derived from the twin–twin intersection always deviates from any possible twinning direction of the obstacle twin. Thus, it is unfavorable for twin nucleation even if the stress concentration increases at the twin–twin intersection. Instead, it activates a basal slip band at the intersection in the obstacle twin. Moreover, the dislocations tend to glide on the twin boundary, where dislocation motion is quicker than in the matrix without the suppression of image force [152]. The increase of applied stress can be relaxed by the twin thickening due to the motions of these dislocations.

On the other hand, the stress state dominates the twin thickening and coalesce after the twin–twin intersection. This is supported by the fact that the twin variants with the highest Schmid factors, or in other words, imposed by the maximum resolve shear stress, can grow fastest and consume the other twin variants [155].

## 5. Applications and Characterization Techniques

### 5.1. The Boundary Strengthening of Bulk Metals

For most bulk metals, tuning the microstructure to effectively impede dislocation motion is the typical method to improve strength. For example, the deformation and strengthening mechanisms of microstructural refinement by deformation at ambient temperatures is owing to the increase of grain and twin boundary densities as well as the increase of boundary misorientation according to the boundary strengthening [156,157,158] and dislocation strengthening [159]. Besides, the concept of precipitation strengthening is derived from the strong impediment of the phase interface to dislocation motion [160].

In addition to the usual methods to improve the mechanical properties of bulk metals, some approaches have been developed according to the other findings of dislocation–boundary interactions. For example, the concept of grain boundary engineering (GBE) by increasing the fraction of special or low Σ CSL boundaries of metals is based on the different structures and energies from random high angle boundaries [23,24,25]. On the other hand, the mechanical properties of Mg alloy has been improved by transition of deformation mechanism from twinning and <a> slip to <c + a> slip [161,162]. This is because the <c + a> slip system of Mg alloy has more independent slip directions than those of <a> slip system, which reduce the stress concentration induced by the dislocation–boundary interaction and increase its plasticity. Therefore, the knowledge of dislocation–boundary interaction is significant not only for fundamental science, but also for alloy/processing design strategies in the development of advanced bulk metals.

### 5.2. The Analytical Descriptions of Boundary Strengthening

Although massive researches show that there exists various kinds of dislocation–boundary interactions, it is still difficult to directly correlate these investigations to the mechanical properties of bulk samples due to the different kinds of restrictions such as the limitation of experimental techniques as summarized afterwards, the internal-microstructure and external-sample size, length-scale effect and the strain rate effect, etc. Hereby, the analytical descriptions of strengthening based on the dislocation–boundary interaction is briefly summarized in the following contents.

#### 5.2.1. Hall–Petch Relation

The boundaries are barriers to dislocation motions. Consequently, in the coarse-grained polycrystalline metals, the strength increases as the grain size decreases [100]. The Hall–Petch relation is based on this rule and is expressed as the following:(10)$$\sigma_{y}{=\;\sigma }_{0}{\;+\;k}_{\mathrm{HP}}D_{\mathrm{av}}^{-1/2}$$
where *σ*_y_ is the yield stress, *D*_av_ is the average grain size, *σ*_0_ and *k*_HP_ are constants. *σ*_0_ is the friction stress, which is the flow stress of an undeformed single crystal or approximately the yield stress of a very coarse-grained polycrystal without texture [99]. The Hall–Petch relation is the simplest analytical description of strengthening derived from dislocation–boundary interactions, which has a great impact on the other analytical descriptions.

Hansen [99] points out that the strengthening should not only come from grain boundaries but also dislocation accumulations in the grain interior, i.e., the dislocation boundaries (Section 2.2). Since the IDBs are assumed to be penetrable to slip and contribute via the forest hardening. Meanwhile the GNBs are assumed to be barriers to slip and contribute by the Hall–Petch strengthening [163,164]. The flow stress is the sum of friction stress, forest and Hall–Petch hardening, which can be expressed as:(11)$$\sigma_{s}{=\;\sigma }_{0}\;+\;\left[k_{\mathrm{HP}}\sqrt{f_{\mathrm{GNB}}}\;+\;M\alpha \mu \sqrt{{3b\theta }_{\mathrm{IDB}}{(1\;-\;f}_{\mathrm{GNB}})}\right]D_{\mathrm{av}}^{-1/2}$$
where *f*_GNB_ is the fraction of GNBs, *M* is the Taylor factor, *α* is a constant, *μ* is the shear modulus, *b* is the Burgers vector and *θ*_IDB_ is the average misorientation angle of IDBs. Since most GNBs and IDBs are high angle and low angle boundaries, respectively, the Equation (11) can be also expressed as:(12)$$\sigma_{s}{=\;\sigma }_{0}\;+\;\left[k_{\mathrm{HP}}\sqrt{f_{\mathrm{HAB}}}\;+\;M\alpha \mu \sqrt{{3b\theta }_{\mathrm{LAB}}{(1\;-\;f}_{\mathrm{HAB}})}\right]D_{\mathrm{av}}^{-1/2}$$
here the subscripts “HAB” and “LAB” refer to the high angle and low angle boundary, respectively.

For the nanocrystalline metals with grain sizes smaller than 20 nm, usually, the dominant deformation mechanism switches from a dislocation-mediated process to the grain boundary sliding. However, Zhou et al. [100] has prevented grain boundary sliding and enhanced Hall–Petch strengthening in the nanocrystalline by relaxation and element segregation. They found that the Hall–Petch strengthening in nanocrystalline with grain size of 3 nm were different from coarse-grained polycrystals: the partial dislocation hardening was as important as full dislocation hardening. Therefore, they proposed a modified Hall–Petch relation as follow:(13)$$\sigma_{y}{=\;\sigma }_{0}\;+\;\frac{k_{\mathrm{HP}}}{\sqrt{D_{\mathrm{av}}}}\;+\;\frac{k_{\mathrm{partial}}}{D_{\mathrm{av}}}$$
where *k*_partial_ is a constant. The third term represents the contribution from partial dislocations, which is inversely proportional to grain size. Meanwhile, Zhang et al. [125,126] have also observed that the dislocation-based plasticity still exists in the sub-20 nm lamellar structures of pearlite steel wires, where the cementite lamellae dissolve at large strains and stabilize the ferrite lamellar boundaries for further deformation.

#### 5.2.2. Thermal Activation Theory

The thermal activation theory is established based on the thermal and athermal nature of flow stress, where there are three types of barriers to dislocation motions: short range, long range, and drag components [165,166]. The short-range component originates from lattice and point defects in the grain interior, whose dimension is about 10 atomic diameters. Since the stress contributed by short range component is influenced by strain rate and temperature, it is absolutely a thermal stress *σ*_TH_. The stress of long-range component is derived from forest dislocations and grain boundaries. This stress is not affected by strain rate and is almost independent of temperature, which is usually regard as an athermal stress *σ*_A_. The stress contribution from drag component is considered to be small when the strain rate is smaller than 10^5^–10^7^ s^−1^. In most cases, the flow stress can be expressed as [166]:*σ*_s_ = *σ*_A_ + *σ*_TH_(14)

Since the athermal stress *σ*_A_ is contributed by forest dislocations and grain boundaries [167,168], it can be expressed as:(15)$$\sigma_{A}=\;M\alpha \mu b\sqrt{\rho }{\;+\;k}_{\mathrm{HP}}D_{\mathrm{av}}^{-1/2}$$
here *ρ* is the dislocation density. The two terms are contributions from forest and Hall–Petch strengthening, respectively. Visser and Ghonem [165] have taken grain boundaries as well as twin boundaries into account by considering the twin boundaries as a decrease of original grain size. The thermal stress is written as:(16)$$\sigma_{A}{=\;\sigma }_{F}\varepsilon^{i}{\;+\;\sigma }_{T}$$
where ε is the strain, *σ*_F_ and *i* are hardening parameters due to the forest strengthening. *σ*_T_ is the stress contributions of twin boundaries, which is expressed as:(17)$$\sigma_{T}{\;=\;\sigma }_{0}{\;+\;\beta F}^{j}$$
where *β* and *j* are constant, *F* is the twin volume fraction.

#### 5.2.3. Precipitation Strengthening

The analytical descriptions change according to the morphologies and the dimensions of phases [169]. In terms of the nano-scale particles (situation represented in the Section 3.5.1), if the particles have coherent interface, are small enough and can be sheared by the dislocations, the stress contribution is written as:(18)$$\sigma_{P1}{\;=\;C}_{1}A_{\mathrm{eff}}^{n}\frac{R^{n}}{L\;+\;2R}$$
here *C*_1_ and *n* are constants, *A*_eff_ is the effective obstacle area, *R* is the radius of particle and *L* is the spacing between two adjacent phases. On the other hand, when the interfaces of particles are incoherent or the sizes of particles are relatively large, where the dislocations and the phase interfaces can only interact by looping (situation represented in the Section 3.5.1), the stress contribution is written as:(19)$$\sigma_{P2}{\;=\;C}_{2}\frac{\mu b}{L}$$
where *C*_2_ is constant. In the case of the phases with dimensions at micro-scale (situations represented in the Section 3.5.1 and Section 3.5.2), e.g., the micro-scale particles and the lamellae, the stress contribution is written as:(20)$$\sigma_{P3}\;=\;\frac{C_{3}}{\sqrt{L}}$$
here *C*_3_ is constant.

### 5.3. Characterization Techniques for Investigating Dislocation–Boundary Interaction

To investigate the strengthening mechanisms and the relationships between the mechanical properties of bulk metals and their dislocation–boundary interactions during deformation, the ideal characterization method is the dynamic observation of the deformation process and interaction behaviors at the full-length scales from nano to macro in three dimensions. Compared with ordinary static experiments, the dynamic observation by in-situ characterization techniques provides the real-time evolution and variation of structural data during the whole interaction process between dislocations and boundaries for further analysis and also provides an insight into the unknown field not obtainable in the post mortem results [12,13,16,17,18,114,151,155]. However, the ideal condition cannot be met by any single in-situ characterization technique due to the limitations on spatial resolution, states and dimensions of specimens, and deformation strains and strain rates. The in-situ techniques are compared below with a focus on SEM, TEM and XRD techniques.

The in-situ SEM technique can produce a high-quality image with a spatial resolution smaller than 1 nm and provide abundant information on microstructure, crystal orientations, phases, Schmid factors, and strains. It has been considered as one of the best methods for the investigation of the twinning behavior during deformation induced by the external stress [149,151,155]. Compared with the in-situ TEM technique, the in-situ SEM technique is capable for investigating the crystal orientation, microstructure and dynamic behavior at both large stains and large scales, which approaches the situation of macro-mechanical properties. However, it cannot give a detailed structure of a boundary as well as dislocation dissociation during twin-boundary interactions. These details need to be further verified by the TEM technique or simulation studies. On the other hand, it is difficult to observe individual dislocations interacting with boundaries by the SEM technique. Even so, multiple findings are obtained in observation of bulk metals coupled with mechanical testing, which are unlikely to be obtained by the TEM technique [15,18,19,20,170]. Therefore, it is promising to investigate dislocation–boundary interactions in bulk metals by in-situ SEM technique at large strains and large scales together with the post mortem observations by TEM.

Since it is difficult to observe individual dislocation interactions by other methods, the in-situ TEM technique has remarkable advantages in characterizing dislocation–boundary interactions due to its high spatial resolution. It facilitates the atomic-scale dynamic observation of dislocation–boundary interactions [12,16]. Additionally, details of dislocation motions during the twin–twin interaction can be revealed by the in-situ TEM technique [13,152]. Moreover, the TEM technique also provides abundant information including atomic number, crystal structure and orientation. Although it seems that the TEM technique meets all the needs for materials research, there are some drawbacks [50,171,172]:The specimen requires a thin foil with thickness typically below 300 nm, which cannot reflect a macroscopic performance of metals;The useful field of view is small, which is hard to reach a large quantity of statistics;Mechanical damage and relaxations of stored deformation may be induced during specimen preparation due to its large free surfaces;It is hard to control imaging and deformation conditions to obtain a qualified field of view during the in-situ experiment.

Consequently, it is hard to relate the data by in-situ TEM technique to the macroscopic mechanical properties of metals.

Compared with the in-situ SEM and TEM techniques, the in-situ XRD technique exhibits many advantages:Non-destructive measurements [173,174], which can be applied on bulk specimen tested by the conventional mechanical equipment rather than pillars or thin foils;A large detecting area up to macro-scale [173,175], which facilitates the research on average behaviors of multiple grains;The information obtained by penetrating several millimeters of bulk crystal volume instead of data from the surface (SEM) or the thin foil (TEM) [174], which is able to build 3D mapping and investigate 3D interactions between dislocations and boundaries [95,176,177];It is more sensitive to elastic strains, which is favorable to study the stress distribution before plastic deformation [178];It is capable to perform experiments at high temperatures (above 900 °C) [179].

Even though the in-situ XRD technique is powerful, some drawbacks have limited applying this method in the studies of interactions between dislocations and boundaries. At first, all information obtained by XRD is based on the analysis of Laue diffraction pattern, which is different from the straightforward results obtained by other microscopic techniques [174]. The most important reason for few research on interactions between dislocations and boundaries is the resolution of the in-situ XRD, which is only at the micro-scale [178,180,181].

## 6. Summary and Outlook

So far, there is still a gap between the knowledge of dislocation–boundary interactions and macroscopic mechanical properties due to many influencing factors and differences between experimental and normal testing conditions. Moreover, there are still uncovered interaction mechanisms which need intensive and comprehensive studies to better understand the nature of dislocation–boundary interactions.

### 6.1. Investigations of Unknown Interaction Mechanisms

#### 6.1.1. Reveal New Mechanisms during Dislocation–Boundary Interaction

Although great progress has been achieved in understanding the dislocation–boundary interactions, new findings are emerging continuously. For example, in the dislocation–twin boundary interaction of nanotwinned Ag, the resistances of dislocation transmission across the nanotwin boundaries are relatively weak, which are not considered as the dominant reason for their high strength. The governing strengthening mechanism of nanotwinned Ag is that the small twin spacing of nanotwinned structure remarkably increases the local curvature of partial dislocations, which requires larger stresses to emerge the partial dislocations into the perfect screw dislocations before transmitting across twin boundaries [18]. Further researches need to reveal the unknown interaction mechanisms in particular for nanocrystalline and nanotwinned metals.

#### 6.1.2. Discover the Origins of Boundary Resistance

The strengthening induced by dislocation–boundary interactions can originate from image forces, boundary absorptions and interactions, and accommodations to the adjacent grains. However, in most cases, the resistances of boundaries are regarded as a whole, and their strengthening contributions have not been clarified in detail. For example, Malyar et al. [17] found that the dislocation–grain boundary interaction shows different behaviors at different strain rates: the dislocations can easily transmit across a grain boundary at a low strain rate while are piled up in front of a grain boundary at high strain rates. This is due to varied difficulty levels of dislocation reorientation with strain rates when the dislocations are emitted into the adjacent grain. Since the resistances of boundaries at different stages of dislocation–boundary interactions may change according to the deformation status, it is important to distinguish their strengthening contributions from each other.

#### 6.1.3. Validate Assumptions and Simulation Results by Experiments

For example, several models and simulations have been performed to explain the dislocation transmission mechanism across the ferrite–cementite interface in pearlitic steels [22,30,122,131,132,133,134]. However, the direct dislocation transmission across the interface has not been confirmed through experimental observations. This is ascribed to the fact that the observation of dislocations in cementite lamellae is relatively difficult by TEM. There are many cases remained at the stage of theory assumptions due to experimental difficulties, which are expected to be solved in future by advanced in-situ techniques.

### 6.2. Comprehensive Studies of Influencing Factors

Directly correlating the practical mechanicalproperties of metals with dislocation–boundary interaction models faces huge challenges. This is because there are numerous influencing factors of dislocation–boundary interactions and none of them can be considered as absolutely independent regardless of others. However, most researches focus on one factor. In this case, the obtained conclusions isolated to other influencing factors may be diverse from case to case and sometimes result in contradictions between different results. Moreover, deformation of polycrystalline metals is the dominant processing of structural metals. These conditions are different from most of the reported results in bi-crystals at relatively low strains and strains rate by TEM and SEM.

So far, the Hall–Petch relation is a model concerning the dislocation–boundary interactions that has been widely recognized as feasible to the practical mechanical properties of metallic materials [99,100]. However, even in this successful case, the grain size is the only factor considered in this relationship, regardless of important factors such as Bravais lattices, boundary structure, misorientation or geometrical condition, dislocation types, strain, strain rate and so on. Also, the other strengthening models on the dislocation–boundary interaction face the same situation as the Hall–Petch relation. However, the experimental results have verified the importance of influencing factors, which are not negligible in the dislocation–boundary interaction. For example, the dislocation types have been confirmed to show distinct behaviors when interacting with twin boundary [31,40,104,105,107] and grain boundary [43], respectively. Additionally, although the resistances to dislocation motions of low-energy and low Σ CSL boundaries are a controversial issue [35,37], they have been applied as grain boundary engineering (GBE) and improve the mechanical properties of metals [23,24,25].

On the other hand, compared with the dislocation–boundary interactions, there is few studies on the influencing factors of twin–boundary interactions. For example, the studies of twin–grain boundary interaction mainly focus on the grain boundary misorientation in hcp metals [14,32,36,149,150]. Whereas most researches of twin–twin interaction investigate what occurs during the interaction [13,38,39,148], but few of them involve investigations of influencing factors. Therefore, there is still a long way to go in the study of dislocation–boundary interactions. To better investigate the behaviors during dislocation–boundary interactions, more experimental data by combining different in-situ testing methods is needed to take cross-validations at different length scales, strains, and strain rates etc., as well as the correlation of these data to the mechanical properties of bulk metals.

## Figures and Tables

**Figure 1 materials-14-01012-f001:**
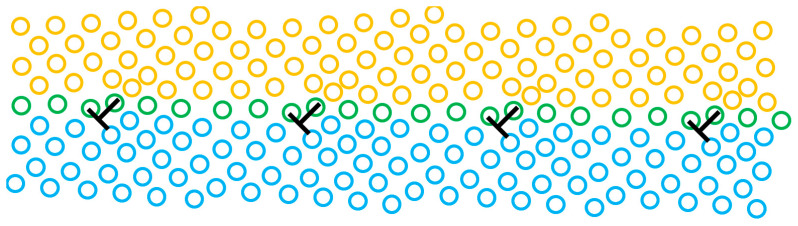
An illustration of a low angle grain boundary.

**Figure 2 materials-14-01012-f002:**
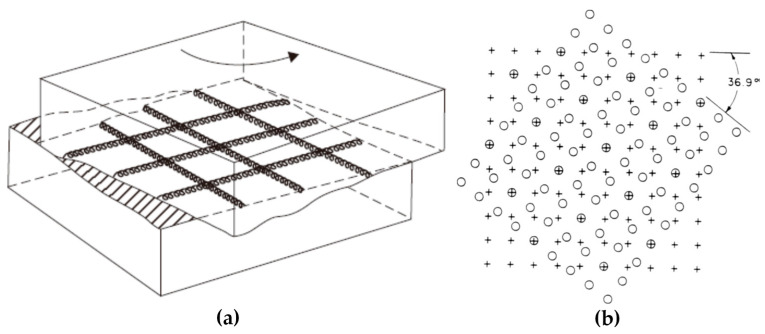
Illustrations of twist grain boundaries. (**a**) Two crystals rotate around the common grain boundary plane; (**b**) Coincident positions of two crystals. (Reprinted from References [26,58], with permission from Elsevier).

**Figure 3 materials-14-01012-f003:**
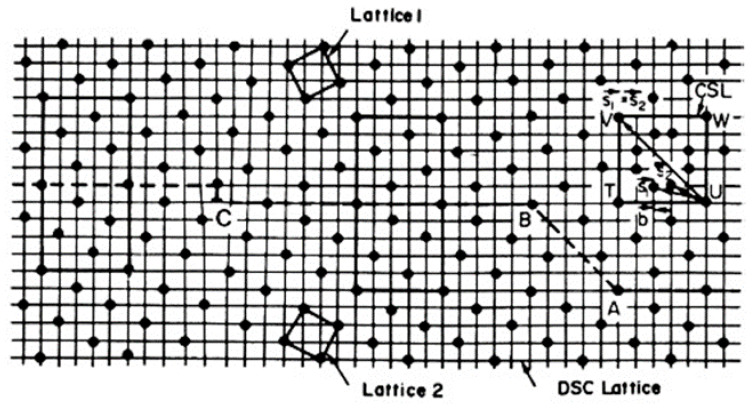
An illustration of CSL lattice and DSC lattice in a grain boundary with CSL lattices of Σ = 5 by a 36.9° rotation around [001] direction. (Reprinted from Reference [65], with permission from Elsevier).

**Figure 4 materials-14-01012-f004:**
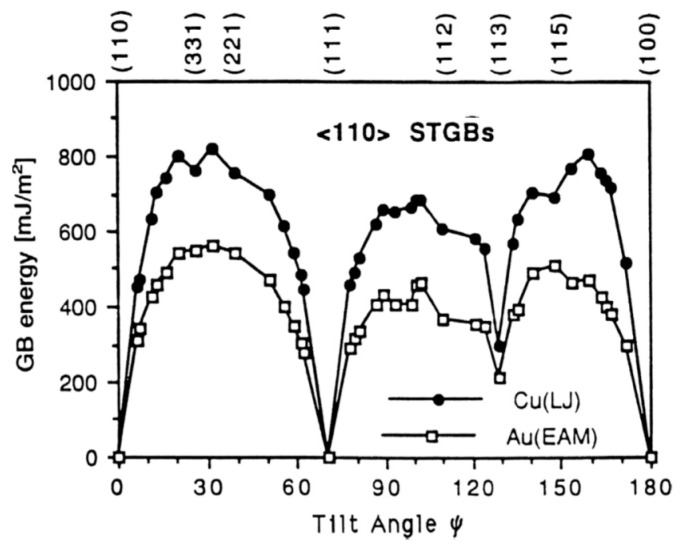
The correlation between grain boundary energies and misorientation angles of <110> symmetrical tilt boundaries in fcc metals (Cu and Au). (Reprinted from Reference [26], with permission from Elsevier).

**Figure 5 materials-14-01012-f005:**
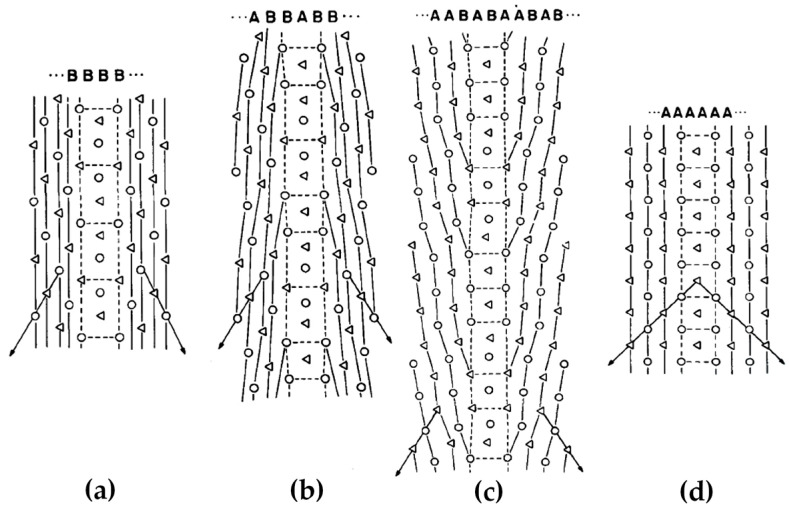
Illustrations of perfect Σ = 5, 17, 37 and 1 grain boundaries from (**a**–**d**), respectively. (Reprinted from Reference [26], with permission from Elsevier)

**Figure 6 materials-14-01012-f006:**
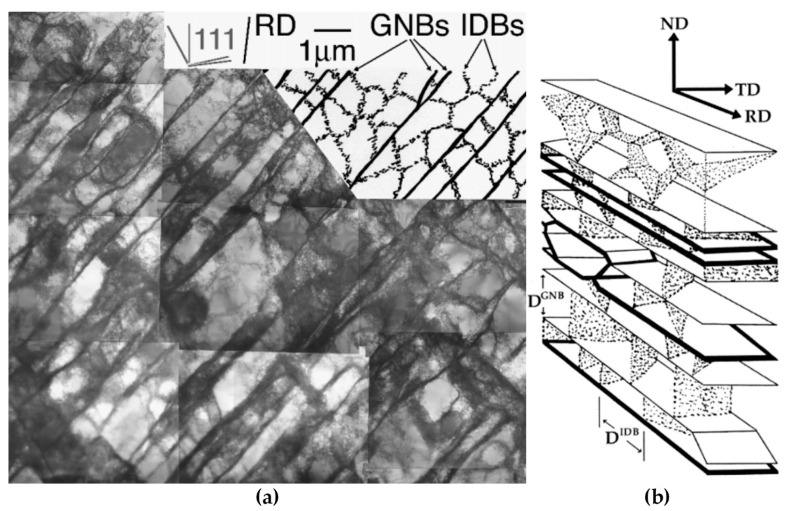
TEM image and the illustration of IDBs and GNBs. (**a**) A TEM image of pure Ni cold rolled to a reduction of 20%; (**b**) An illustration of IDBs and GNBs. (Reprinted from References [73,74], with permission from Elsevier).

**Figure 7 materials-14-01012-f007:**
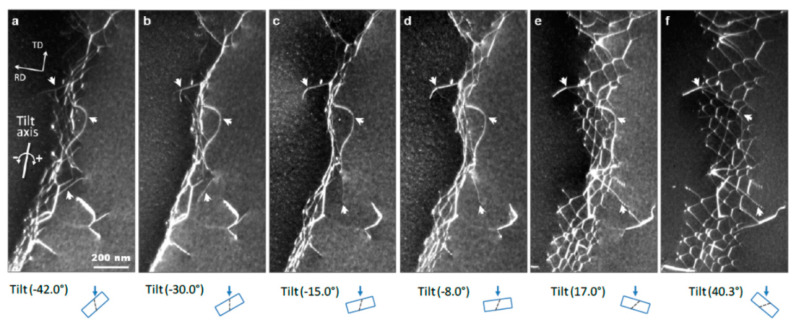
TEM images of a GNB observed at different tilting angles to incident electron beams. (**a**–**f**) show the images with tilting angles of −42.0°, −30°, −15°, −8°, 17° and 40.3°, respectively (Reprinted from Reference [77], with permission from Chuanshi Hong).

**Figure 8 materials-14-01012-f008:**
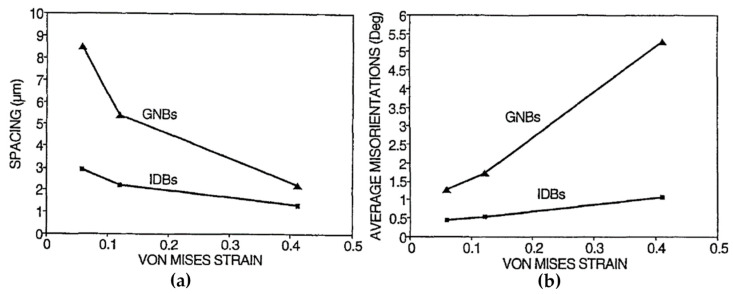
The variations of (**a**) boundary spacing and (**b**) misorientation angle of IDBs and GNBs in dependence of strain in cold-rolled pure Al, respectively. (Reprinted from Reference [79], with permission from Elsevier).

**Figure 9 materials-14-01012-f009:**
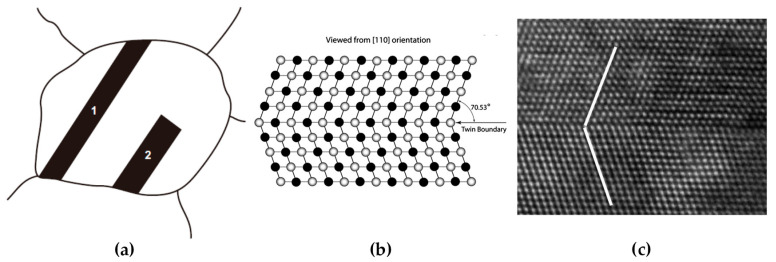
(**a**) A schematic diagram of coherent and incoherent twin boundaries; (**b**) A mirror symmetry of atoms in a coherent twin boundary in fcc metals viewed from the [110] direction; (**c**) A typical HREM image of a twin in a fcc metal. (Reprinted from References [27,58], with permission from Elsevier).

**Figure 10 materials-14-01012-f010:**
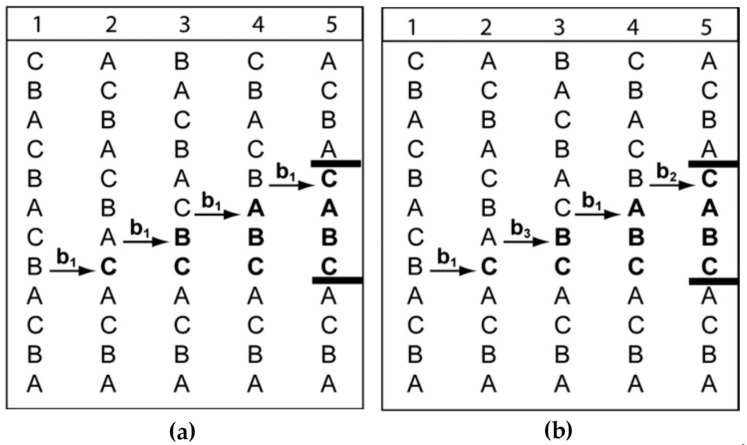
The step-by-step process of producing a four-layer deformation twin by the slip of (**a**) identical and (**b**) different partial dislocations on successive slip planes [27].

**Figure 11 materials-14-01012-f011:**
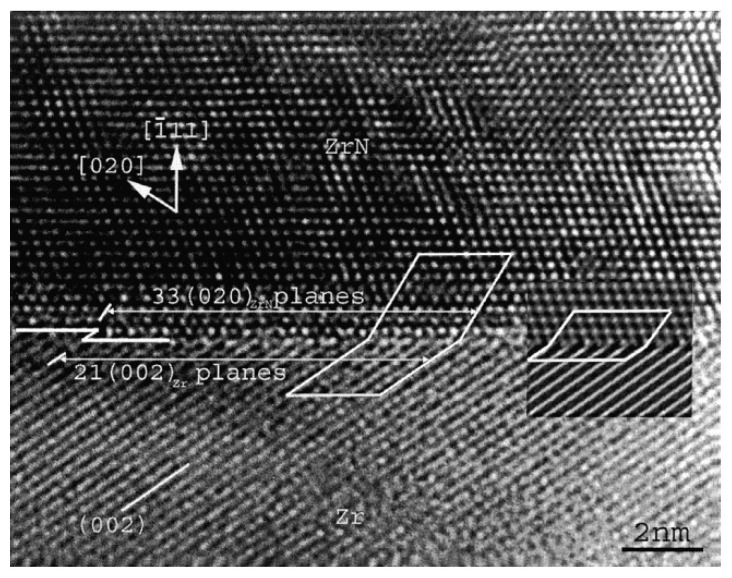
HREM image of an incoherent Zr-ZrN interface in a Zr-N alloy. (Reprinted from Reference [87], with permission from Elsevier).

**Figure 12 materials-14-01012-f012:**
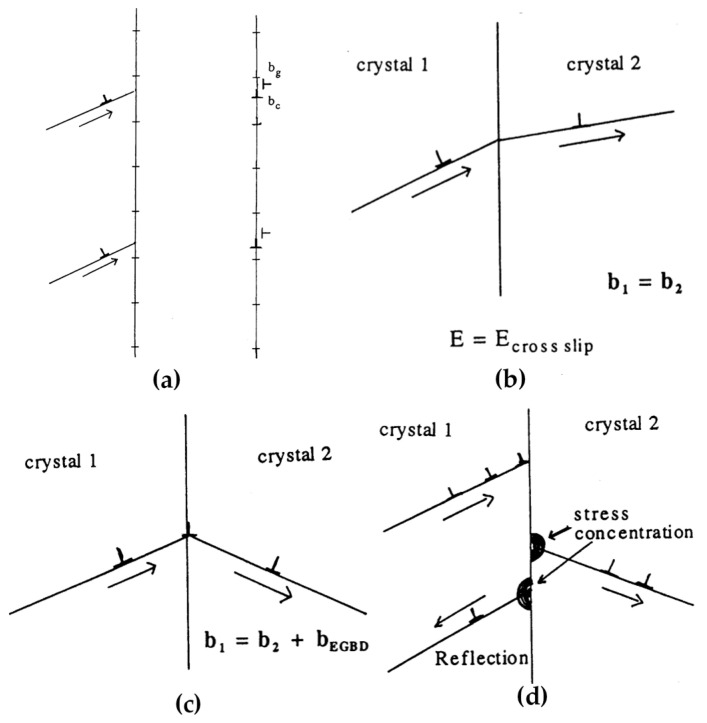
Three types of interactions between dislocations and grain boundaries. The schematic illustrations of (**a**) absorption; (**b**) transmission without residual dislocation; (**c**) transmission leaving residual dislocation; (**d**) reflection, respectively. (Reprinted from Reference [33], with permission from Elsevier).

**Figure 13 materials-14-01012-f013:**
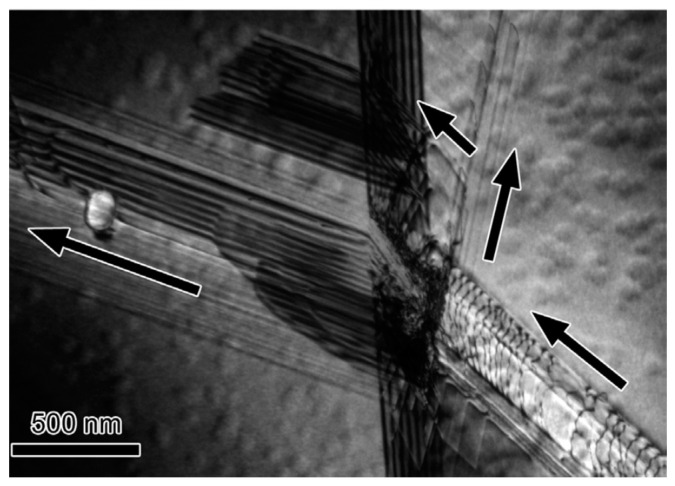
TEM image of the dislocation–grain boundary interaction with direct and indirect transmission and reflection. (Reprinted from Reference [93], with permission from Elsevier).

**Figure 14 materials-14-01012-f014:**
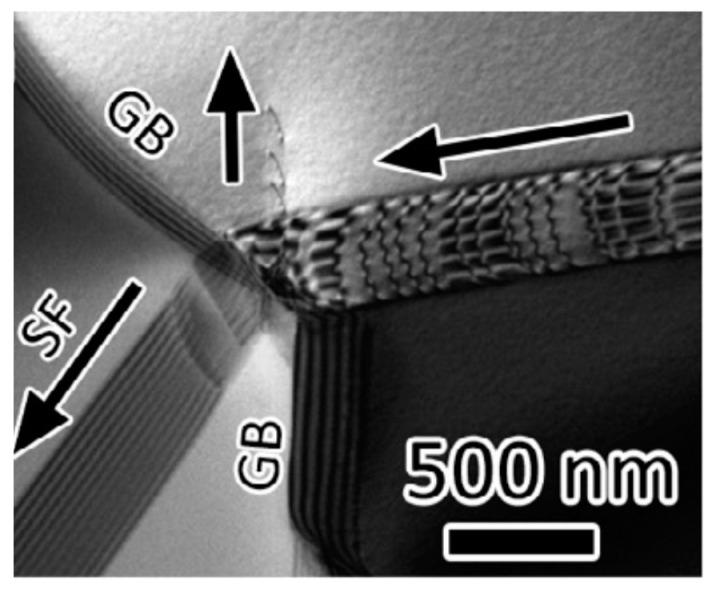
TEM image of twin formation induced by the dislocation–grain boundary interaction. (Reprinted from Reference [29], with permission from Elsevier).

**Figure 15 materials-14-01012-f015:**
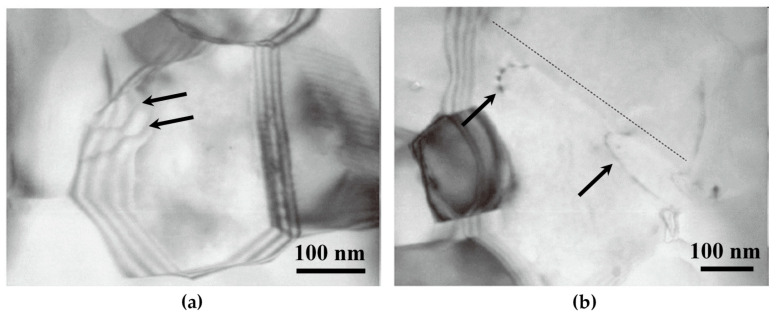
The dislocation–grain boundary interaction in an ultrafine-grained Al at different strains. (**a**) The dislocations move inside a grain boundary at a low strain; (**b**) The dislocations emit into the grain interior at a relatively high strain. (Reprinted from Reference [42], with permission from Elsevier).

**Figure 16 materials-14-01012-f016:**
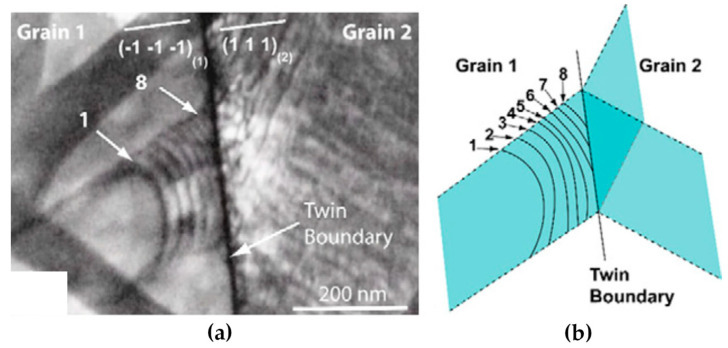
The process of dislocation transmission across a coherent twin boundary. (**a**) TEM image and (**b**) sketch showing dislocation pile-ups in front of a twin boundary, and the dislocation transmission by direct and indirect modes. (Reprinted from Reference [101], with permission from Elsevier).

**Figure 17 materials-14-01012-f017:**
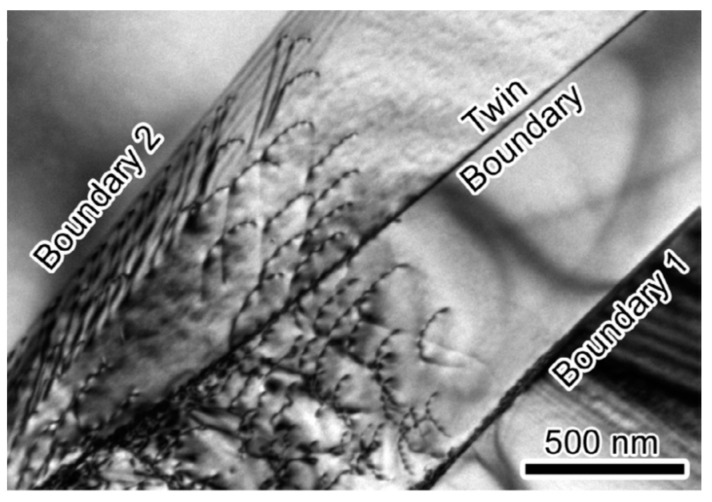
TEM image of direct dislocation transmission without leaving residual dislocations on the coherent twin boundary. (Reprinted from Reference [29], with permission from Elsevier).

**Figure 18 materials-14-01012-f018:**
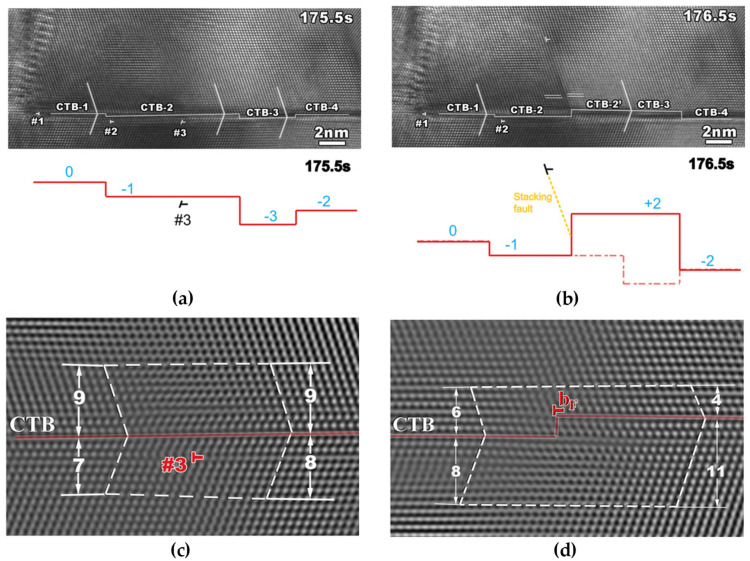
HREM images and illustrations of dislocation transmission leaving residual dislocations and forming steps on the coherent twin boundary. (**a**,**c**) Images and illustration before transmission; (**b**,**d**) Images and illustration after transmission. (Reprinted from Reference [106], with permission from Elsevier).

**Figure 19 materials-14-01012-f019:**
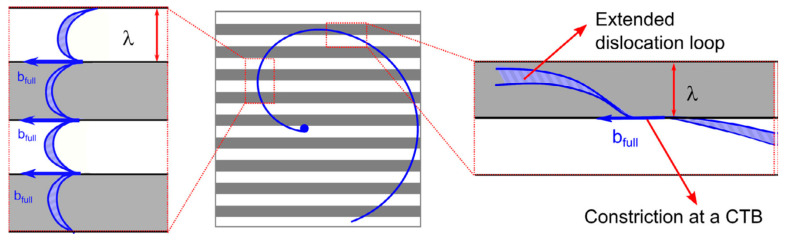
Schematic of the local curvature of partial dislocations between two adjacent coherent twin boundaries during dislocation transmission by the cross-slip mode. (Reprinted from Reference [18], with permission from Elsevier).

**Figure 20 materials-14-01012-f020:**
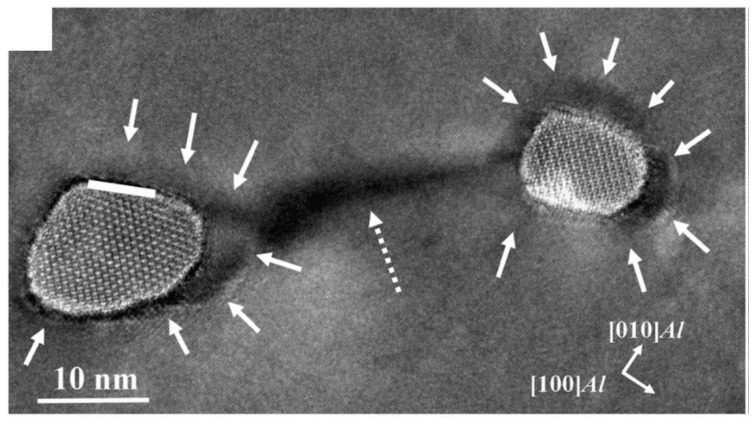
TEM image of looping interactions around particles. (Reprinted from Reference [21], with permission from Elsevier).

**Figure 21 materials-14-01012-f021:**
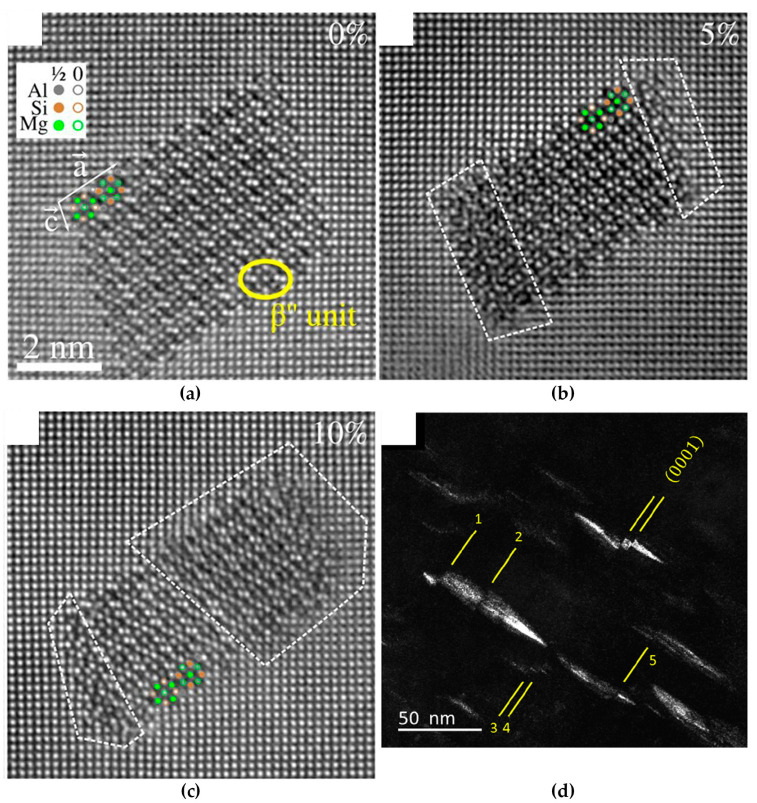
TEM images of shearing interactions. (**a**–**c**) Images of a particle partially sheared by dislocations at compressive engineering strains of 0%, 5% and 10%, respectively; (**d**) Image of particles sheared by arrays of dislocations. (Reprinted from References [114,117], with permission from Springer Nature and Elsevier).

**Figure 22 materials-14-01012-f022:**
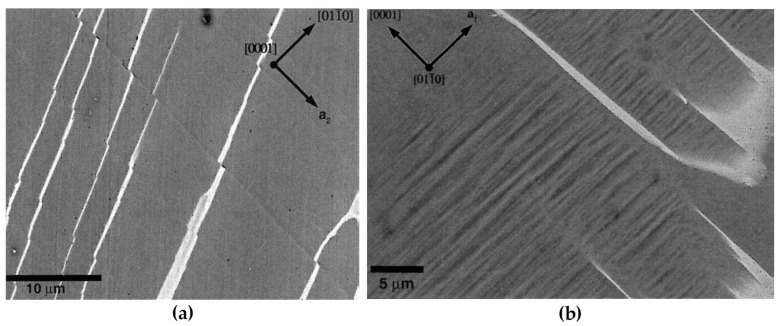
SEM images of interaction between slip and lamellar interfaces. (**a**) The prism slip; (**b**) The basal slip. (Reprinted from Reference [49], with permission from Elsevier).

**Figure 23 materials-14-01012-f023:**
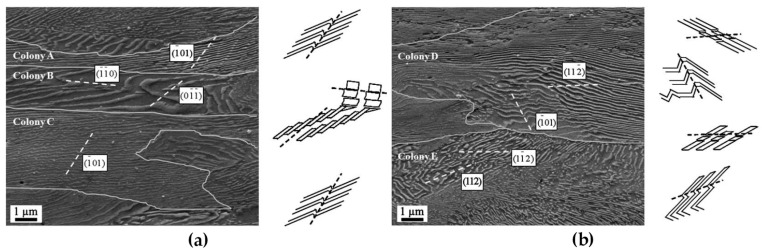
SEM micrographs showing the slip transfer of pearlite by fracture or bending of cementite lamellae. (**a**) {110} slip transfers; (**b**) {112} slip transfers. (Reprinted from Reference [10], with permission from Elsevier).

**Figure 24 materials-14-01012-f024:**
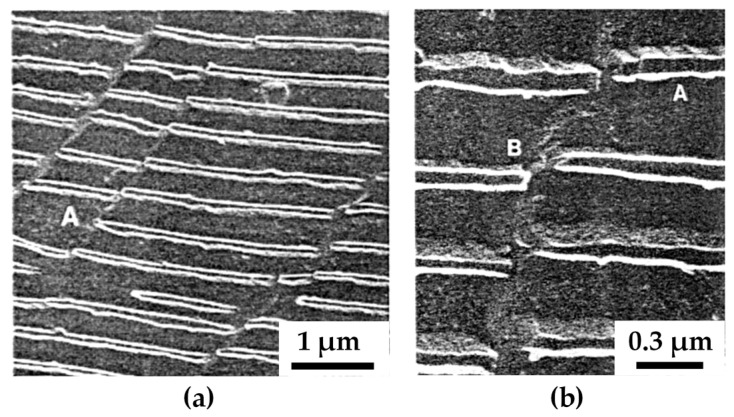
SEM micrographs showing the fracture of cementite lamellae caused by slip transfer. (**a**) and (**b**) are the SEM images at different magnifications. (Reprinted from Reference [22], with permission from Elsevier).

**Figure 25 materials-14-01012-f025:**
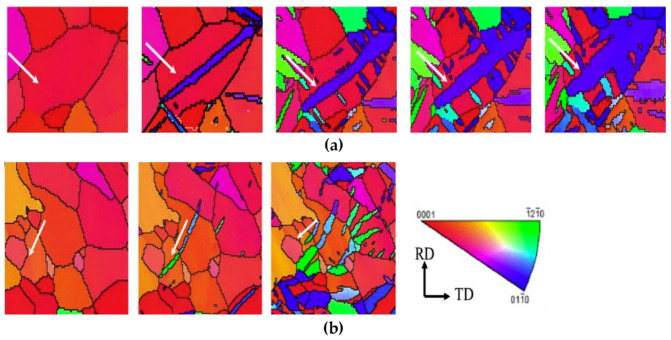
The process of twin transmission investigated by in-situ SEM mechanical testing with compressions strains of (**a**) 0, 1.6%, 3.1%, 4.2%, 5.4% and (**b**) 0%, 1%, 2%, respectively. (Reprinted from Reference [151], with permission from Elsevier).

**Figure 26 materials-14-01012-f026:**
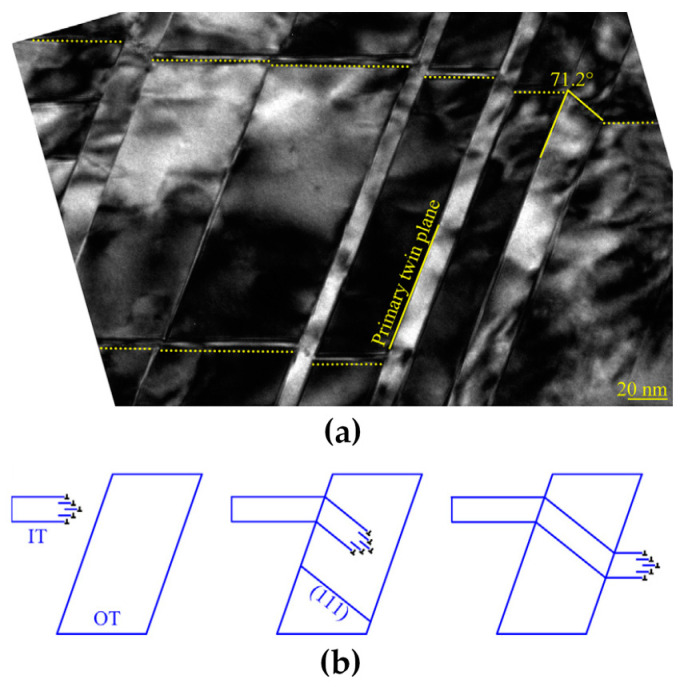
The ordinary twin transmission across twin boundaries in bcc and fcc metals. (**a**) The TEM image; (**b**) The schematic illustration showing the process of twin intersection. (Reprinted from Reference [39], with permission from Elsevier).

**Figure 27 materials-14-01012-f027:**
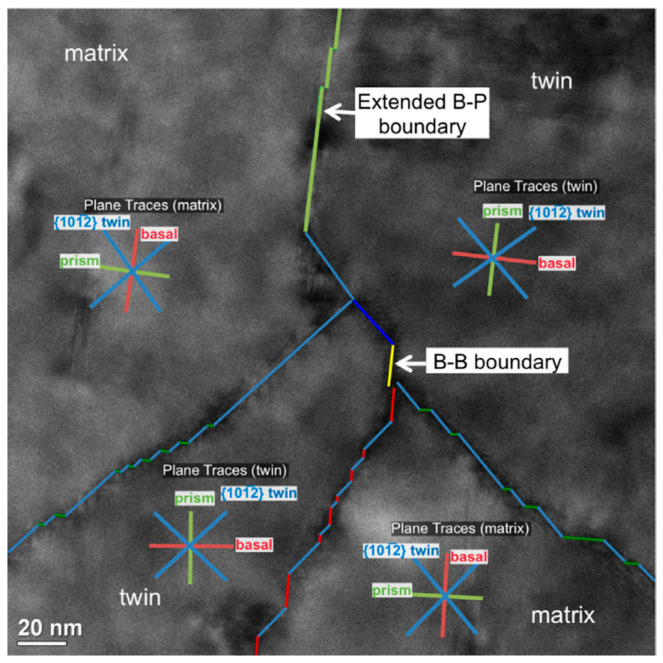
TEM image of a twin–twin intersection with a B-B boundary in Mg. (Reprinted from Reference [13], with permission from Elsevier).

**Figure 28 materials-14-01012-f028:**
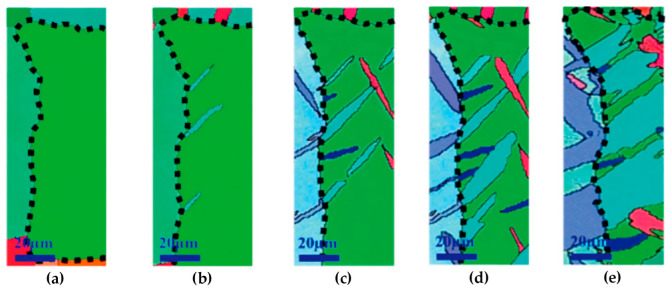
The twin–twin interaction in Mg at true strains of (**a**) 0%, (**b**) 1.2%, (**c**) 2.5%, (**d**) 5.5% and (**e**) 8%, respectively. (Reprinted from Reference [155], with permission from Elsevier).

**Figure 29 materials-14-01012-f029:**
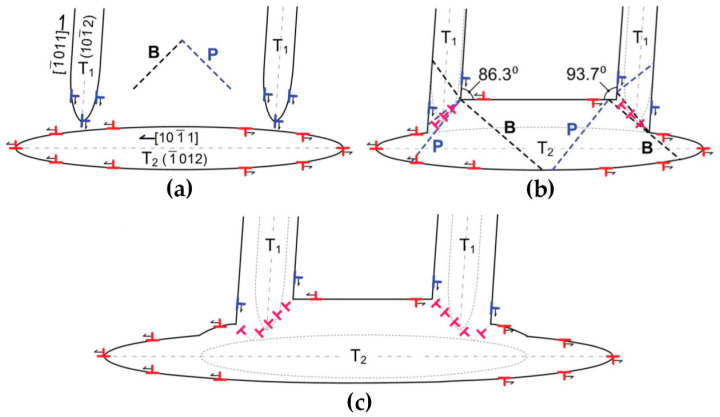
Schematic illustrations of a twin–twin intersection. (**a**) Before impingement; (**b**) Formation of a twin–twin intersection; (**c**) Twin growth after the twin–twin intersection. (Reprinted from Reference [153], with permission from Taylor & Francis).

## Data Availability

No new data were created or analyzed in this study. Data sharing is not applicable to this article.

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
