# Peer review of "Interactions between Dislocations and Boundaries during Deformation"

_materials, 2021, doi:10.3390/ma14041012_

Round 1
Reviewer 1 Report
The manuscript has provided a good survey of the reported researches in the literature. The content is suitable for beginner to intermediate audiences. Many seminal and original papers which dealt with theory of dislocation activities and its interaction with obstacles have not been included (e.g. Kocks, Ashby).The summery is more qualitative and descriptive. It is expected to provide mathematical expression and analytical description that have been developed and found in the literature for these dislocation-boundary interactions. Further, the main implications of dislocation-boundary interaction for strengthening has not been discussed. Again, abundant analytical descriptions of strengthening in terms of dislocation-boundary interaction are available in the literature and its survey is lacking in this manuscript. Another shortcoming is related to thermal and athermal nature of dislocation-boundary interactions which has not been considered in this manuscript. Section 5, the discussion on characterization method is very primitive; it should be either removed or be expanded with more critical survey and examples.
The last section on Outlook and future direction is very qualitative. Either some experiments that cannot be described by existing theory should be included, or some modelling work that requires additional experimental research and verification should be provided.
Overall, as an introductory and basic review the submitted manuscript can be published.
Author Response
Ref. No.: materials-1064961
Title: Interactions of Dislocations and Boundaries during Deformation
Materials
Dear Editor,
Thank you very much for your attention and the reviewer’s comments concerning our manuscript (Ref. No.: materials-1064961). These comments are all valuable and very helpful for revising and improving our paper. We have studied these comments carefully and made a careful correction in the revised manuscript (The revisions are outlined by blue text). The detailed responses to the reviewer's comments are as follows.
Reviewer's comments:
Question #1
Many seminal and original papers which dealt with theory of dislocation activities and its interaction with obstacles have not been included (e.g. Kocks, Ashby).The summery is more qualitative and descriptive. It is expected to provide mathematical expression and analytical description that have been developed and found in the literature for these dislocation-boundary interactions.
Answer: Thanks for your helpful suggestion. We have supplemented the mathematical expressions and analytical descriptions for dislocation-boundary interactions in our topic. Please refer to the Section 3.1.2 on the Page 17 to 18.
On the other hand, the theories by Ashby and Kocks on interactions between dislocations and obstacles are no doubt extraordinary works. It is our honor to read such works and we have learned a lot. After our discussion, we think these works are not so relevant to our topic, which is focusing on the behaviors and processes of dislocation-boundary interactions, rather than the stress fields and dislocation shapes in the vicinity of the intersection [1-3]. Additionally, the topic is focused on the resistances and influences of boundaries against the dislocation motions, where the parameters and differences of boundaries should be highly concerned. It is different from the situations of dislocation multiplications in the constant structure with few effects of boundaries [4-7]. We are sorry that these works are not cited in the revised manuscript. We hope for your understanding about it.
The cited references above are shown as follows:
- Frost, H.J.; Ashby, M.F. Motion of a dislocation acted on by a viscous drag through an array of discrete obstacles. J. Appl. Phys. 1971, 42, 5273–5279.
- Weeks, R.W.; Pati, S.R.; Ashby, M.F.; Barrand, P. The elastic interaction between a straight dislocation and a bubble or a particle. Acta Metall. 1969, 17, 1403–1410.
- Bacon, D.J.; Kocks, U.F.; Scattergood, R.O. The effect of dislocation self-interaction on the orowan stress. Philos. Mag. 1973, 28, 1241–1263.
- Kocks, U.F.; Mecking, H. Physics and phenomenology of strain hardening: The FCC case. Prog. Mater. Sci. 2003, 48, 171–273.
- Regazzon, G.; Kocks, U.F.; Follansbee, P.S. Dislocation kinetics at high strain rates. Acta Metall. 1987, 35, 2865–2875.
- Kocks, U.F.; Mecking, H. Dislocation kinetics at not-so-constant structure. Dislocation Model. Phys. Syst. 1981, 173–192.
- Meckings, H.; Kocks, U.F. Kinetics of flow and strain-hardening. Acta Metall. 1981, 29, 1865–1875.
Question #2
The main implications of dislocation-boundary interaction for strengthening has not been discussed.
Answer: Thanks for your valuable suggestion. We sincerely think this suggestion is a good idea and make the manuscript much better. We have supplemented the contents in the keywords, the first paragraph of Section 1 and the Section 5.1.
Question #3
Abundant analytical descriptions of strengthening in terms of dislocation-boundary interaction are available in the literature and its survey is lacking in this manuscript. Another shortcoming is related to thermal and athermal nature of dislocation-boundary interactions which has not been considered in this manuscript.
Answer: Thanks for your helpful suggestion. Indeed, the absences of analytical descriptions of strengthening and thermal activation theories are not sufficient for a comprehensive review. We are glad to see that the manuscript improved remarkably after supplementing these contents. Please refer to the Section 5.2 and the Page 4.
Question #4
Section 5, the discussion on characterization method is very primitive; it should be either removed or be expanded with more critical survey and examples.
Answer: Thanks for your helpful suggestion. We have revised the section 5 and made it shorten and concise. The revised contents are given in the Section 5.3.
Question #5
The last section on Outlook and future direction is very qualitative. Either some experiments that cannot be described by existing theory should be included, or some modelling work that requires additional experimental research and verification should be provided.
Answer: Thanks very much. We have supplement some examples in the Section 6.2.
It has to be admitted that dislocation-boundary interaction is a very broad field. The main topic of present manuscript is a comprehensive review on existing experimental works of dislocation-boundary interactions. To make the manuscript covered with abundant and detailed information, it requires the experiences and knowledge of many outstanding researchers specialized in different areas of interests. At the moment, it is not likely for us to do so. We hope for your understanding about it.
Thanks again for your careful comments. We hope you would be satisfied with this revision!

Reviewer 2 Report
The paper is a review on the different interaction phenomena between dislocations and crystalline grain and phase boundaries in mostly metallic materials. It summarizes both the effect of dislocation motion on boundaries and change of dislocations during interaction with boundaries from many points of view. Due to the complexity of grain boundaries and therefore the targeted topic, the authors present a broad overview, but presenting only qualitative approaches in their paper.
The text is somewhat hard to read, due to the strange and long expressions. Therefore I suggest polishing of the English language and make extended corrections before publishing.
I recommend to use the word dislocation instead of dislocation activities in the text. This change can make the text simpler (especially in sections 1,2, and 6) for the reader and it will not change the meaning of the sentences.
For example the title may be changed as: Interaction of dislocations and boundaries during deformation.
The authors mention in the abstract and also later in the text that they discuss metals at ambient temperatures. Although, it is possibly true for the cited works, it would be better to refer to a specific temperature range with respect to the melting point (e.g. 0.2<T/Tm<0.5).
Figures 2 and 3 is somewhat misleading, because the plotted grain boundaries can be constructed from two sorts of dislocation series (each series terminates atomic planes at one side of the boundary).
The paper contains repetitions, e.g. similar or same information is disclosed in both sections 5.1 and 5.2. Therefore section 5 can be shortened.
Please check, whether the content listed in section 1 is correct. For example section 3.5 deals with different content as it is indicated here.
Some further simple corrections:
line 49: “The rejection of dislocation of ...”
lines 79-80: “... during dislocation – grain boundary interaction ...”
line 84: “... because dislocations are subjected ...”
lines 475-476: “... observed between two phases ...”
line 502-503: “... increase of misorientation, defect density, change of Bravais latticeand composition between crystals between the sides of the boundary.”
line 544: “... intensive studies for decades as detailed in the following.”
line 546: reference is needed
line 549: It is not clear how can a dislocation rotate when passing though a boundary, without leaving part of the Burgers vector behind.
line 616: “Additionally, some other factors are also reviewed in details as follows.”
line 647: It would be useful to describe LRB somewhere, because it is not widely known.
Line 698: “... some studies show ...”
781-782: Please clarify this sentence
line 801: “At elevated strains, ...”
line 817: The first sentence of this paragraph is a repetition. In the next sentence “The different studies on ...”
line 885: Clarify the sentence.
Line 1020: “..., the largest contribution is found to have the ...”
line 1035: “... ascibed to dislocations ...”
line 1037: “... different Bravais lattice, ...”
Section 3.5: dislocation motion in second phase (in ceramic particles) are ascribed to the low mobility of dislocations in these particles due to the strong covalent bonds and high melting point of the second phase.
Line 1396: “.. large number of dislocations are engaged ...”
line 1443: “... techniques were developped ...”
line 1500: “X-ray diffraction ...”
line 1581: “1500 s-1”
Author Response
Ref. No.: materials-1064961
Title: Interactions of Dislocations and Boundaries during Deformation
Materials
Dear Editor,
Thank you very much for your attention and the reviewer’s comments concerning our manuscript (Ref. No.: materials-1064961). These comments are all valuable and very helpful for revising and improving our paper. We have studied these comments carefully and made a careful correction in the revised manuscript (The revisions are outlined by blue text). The detailed responses to the reviewer's comments are as follows.
Reviewer's comments:
Question #1
The text is somewhat hard to read, due to the strange and long expressions. Therefore I suggest polishing of the English language and make extended corrections before publishing.
Answer: Thanks for your helpful suggestion. We have made corrections and improved the expressions throughout the text. Since we have revised a large amount of them, we do not outline them by blue text. We hope that our revisions can meet with your approval.
Question #2
I recommend to use the word dislocation instead of dislocation activities in the text. This change can make the text simpler (especially in sections 1,2, and 6) for the reader and it will not change the meaning of the sentences.
For example the title may be changed as: Interaction of dislocations and boundaries during deformation.
Answer: Thanks for your careful comment. We sincerely think this suggestion is a good idea and make the manuscript much better. We have changed the expressions “dislocation activities” to be “dislocations”. Please refer it to the title. We do not outline the other changes by blue text due to the concern of better reviewing.
Question #3
The authors mention in the abstract and also later in the text that they discuss metals at ambient temperatures. Although, it is possibly true for the cited works, it would be better to refer to a specific temperature range with respect to the melting point (e.g. 0.2<T/Tm<0.5).
Answer: Thanks very much. We have searched for the corresponding literatures of temperature dependence corresponding to the interactions between boundaries and dislocations. We found a limited number of researches and most of them are investigated by simulations [1-3]. According to these works, the temperature range of dislocation-boundary interactions studied in our manuscript should be at about 300 K to 600 K. If we calculated the temperature range with respect to the melting point, the value should be about 0.15 to 0.31 for Fe (melting point at ~1823 K), while the value is about 0.30 to 0.62 for Mg (melting point at ~923 K). Therefore, it is hard to refer to a specific temperature range with respect to the melting point in our manuscript. Moreover, according to the thermal activation theory, the resistance of dislocation-boundary interaction is an athermal stress, which is independent from temperature and strain rate when the temperature is lower than a critical value [4,5]. However, a recent study shows that the dislocation-grain boundary interactions are different at different strain rates [6]. These factors indicate that the influences of temperature and strain rate are still an unclear area. Therefore, we supplement the description to define the ambient temperatures from about 0 °C to 200 °C in the first sentence of Section 1. We hope for your understanding about it.
The references cited above are given as follows:
- Terentyev, D.; Malerba, L.; Bacon, D.J.; Osetsky, Y. The effect of temperature and strain rate on the interaction between an edge dislocation and an interstitial dislocation loop in α-iron. J. Phys. Condens. Matter 2007, 19.
- Shim, J.H.; Kim, D.I.; Jung, W.S.; Cho, Y.W.; Hong, K.T.; Wirth, B.D. Atomistic study of temperature dependence of interaction between screw dislocation and nanosized bcc Cu precipitate in bcc Fe. J. Appl. Phys. 2008, 104, 1–5.
- Chandra, S.; Samal, M.K.; Chavan, V.M.; Patel, R.J. Atomistic simulations of interaction of edge dislocation with twist grain boundaries in Al-effect of temperature and boundary misorientation. Mater. Sci. Eng. A 2015, 646, 25–32.
- Visser, W.; Ghonem, H. Dynamic flow stress of shock loaded low carbon steel. Mater. Sci. Eng. A 2019, 753, 317–330.
- Conrad, H. Thermally activated deformation of metals. Jom 1964, 16, 582–588.
- Malyar, N. V.; Dehm, G.; Kirchlechner, C. Strain rate dependence of the slip transfer through a penetrable high angle grain boundary in copper. Scr. Mater. 2017, 138, 88–91.
Question #4
Figures 2 and 3 is somewhat misleading, because the plotted grain boundaries can be constructed from two sorts of dislocation series (each series terminates atomic planes at one side of the boundary).
Answer: Thanks for your helpful suggestion. As you said, the structure of grain boundaries are very complicated. The illustrations in our manuscript may be misleading. Consequently, we have removed these pictures and the corresponding descriptions in the Section 2.1. Furthermore, we noted in the manuscript that the grain boundary may be constructed by two sorts of dislocation series on the Page 5.
Question #5
The paper contains repetitions, e.g. similar or same information is disclosed in both sections 5.1 and 5.2. Therefore section 5 can be shortened.
Answer: Thanks for your helpful suggestion. We have revised the Section 5 and made it shorten and concise. The revised contents are given in the Section 5.3.
Question #6
Please check, whether the content listed in section 1 is correct. For example section 3.5 deals with different content as it is indicated here.
Answer: Thanks for your careful comment. We apologize for your misunderstanding due to the unclear descriptions in the Section 1. We have revised the contents corresponding to the Section 3.5 to make it more precise and specific. The detailed revisions are shown on the Page 3.
Question #7
Line 49: “The rejection of dislocation of ...”
Answer: Thanks for your careful comment. We have revised the word “rejection” to be “resistance”. The revision is on the Page 2.
Question #8
Lines 79-80: “... during dislocation-grain boundary interaction ...”
Answer: Thanks very much. We have added the word “interaction” here. The revision is also on the Page 2.
Question #9
Line 84: “... because dislocations are subjected ...”
Answer: Thanks for your careful comment. We have removed the word “motions”. The revision is also on the Page 2.
Question #10
Lines 475-476: “... observed between two phases ...”
Answer: Thanks very much. We have deleted the “in interfaces” here. The revision is on the Page 12.
Question #11
Line 502-503: “... increase of misorientation, defect density, change of Bravais lattice and composition between crystals between the sides of the boundary.”
Answer: Thanks for your helpful suggestion. We have revised the corresponding sentence. Please refer to the sentence on the Page 4.
Question #12
Line 544: “... intensive studies for decades as detailed in the following.”
Answer: Thanks for your helpful suggestion. We have supplemented the content in the corresponding sentence. Please also refer to the sentence on the Page 14.
Question #13
Line 546: reference is needed.
Answer: Thanks very much. We have added the references here. The revision is also on the Page 14.
Question #14
Line 549: It is not clear how can a dislocation rotate when passing though a boundary, without leaving part of the Burgers vector behind.
Answer: Thanks for your helpful suggestion. We must apologize for the unclear description here, which leads to your misunderstanding. We have revised it as the following sentences:
“Considering a dislocation impinges on a grain boundary, the dislocation lines may rotate to be parallel to the grain boundary owing to the image force. Then, the dislocations may dissociate or glide along the grain boundary once impinging on the boundary, which further appears to be absorption, transmission or reflection.”
Please refer to the contents on the Page 14.
On the other hand, the situation that direct transmission without leaving partial dislocation behind has been explained in the first paragraph on the Page 15.
Question #15
Line 616: “Additionally, some other factors are also reviewed in details as follows.”
Answer: Thanks for your careful comment. We have revised the corresponding sentence. The revision is on the Page 17.
Question #16
Line 647: It would be useful to describe LRB somewhere, because it is not widely known.
Answer: Thanks for your helpful suggestion. We have supplemented the expression of LRB. The revision is on the Page 17 and 18.
Question #17
Line 698: “... some studies show ...”
Answer: Thanks for your helpful suggestion. We have removed the word “latest”. The revision is on the Page 19.
Question #18
Line 781-782: Please clarify this sentence
Answer: Thanks very much. We have revised it to be clear and added the reference here:
“The large resistance results in a confined number of dislocation pile-ups and requires larger stress to increase this number. With the decrease of the grain size to nano-scale, the number of pile-up eventually reduces to one.”
The revision is on the Page 21. For more information, please refer to the corresponding contents in the last paragraph on the Page 479 of the cited reference as follow:
- Meyers, M.A.; Mishra, A.; Benson, D.J. Mechanical properties of nanocrystalline materials. Prog. Mater. Sci. 2006, 51, 427–556.
Question #19
Line 801: “At elevated strains, ...”
Answer: Thanks for your careful comment. We have revised the word to be “elevated”. The revision is also on the Page 21.
Question #20
Line 817: The first sentence of this paragraph is a repetition. In the next sentence “The different studies on ...”
Answer: Thanks very much. We must apologize for your misunderstanding of the sentence. We have revised the sentence to be clear:
“As regard to incoherent twin boundaries in coarse-grained metals, their interactions with dislocations show no different behaviors compared with grain boundaries.”
The revision is on the Page 22.
Question #21
Line 885: Clarify the sentence.
Answer: Thanks very much. We have cited the reference of the sentence in the original manuscript. The sentence is currently on the Page 23 of revised manuscript.
For more information, please refer to the cited reference as follow:
- Yang, G.; Ma, S.Y.; Du, K.; Xu, D.S.; Chen, S.; Qi, Y.; Ye, H.Q. Interactions between dislocations and twins in deformed titanium aluminide crystals. J. Mater. Sci. Technol. 2019, 35, 402–408.
The corresponding contents are on the Page 404 and clarified by Fig. 3 of the cited reference.
Question #22
Line 1020: “..., the largest contribution is found to have the ...”
Answer: Thanks for your careful comment. We have revised the sentence and removed the original contents. The revision is on the Page 26.
Question #23
Line 1035: “... ascribed to dislocations ...”
Answer: Thanks for your careful comment. We have revised the sentence and removed the original contents. The revision is also on the Page 27.
Question #24
Line 1037: “... different Bravais lattice, ...”
Answer: Thanks for your careful comment. We have revised the sentence. The revision is also on the Page 27.
Question #25
Section 3.5: dislocation motion in second phase (in ceramic particles) are ascribed to the low mobility of dislocations in these particles due to the strong covalent bonds and high melting point of the second phase.
Answer: Thanks for your helpful suggestion. We have supplement the content and references. The description is as follow:
“Compared with the dislocation mobility in the metallic matrix, the mobility in the non-metallic particles is low due to the strong covalent bonds and high melting point of the particles.”
The revision is on the Page 28.
.
Question #26
Line 1396: “.. large number of dislocations are engaged ...”
Answer: Thanks for your careful comment. We have revised the word “great”. The revision is on the Page 37.
Question #27-29
Line 1443: “... techniques were developed ...”
Line 1500: “X-ray diffraction ...”
Line 1581: “1500 s-1”
Answer: Thanks very much. The Section 5 has been revised. The corresponding sentences have been removed.
Thanks again for your careful comments. We hope you would be satisfied with this revision!

Reviewer 3 Report
This is an excellent review article. Interactions between dislocation activities and boundaries during deformation are fundamental to understand the mechanical properties of alloys and metals. There is no doubt this article should be published in Materials. I am sure it will get a large impact. I only have minor suggestions for the authors.
- At the beginning of the introduction “The mechanical properties of metals at ambient temperature are mainly affected by their microstructures, except for the intrinsic properties determined by their chemical compositions [1,2]. This is because plastic deformation of metals is generated by dislocation activities, including nucleation and gliding of full edge/screw dislocations as well as 28 nucleation and propagation of deformation twins [3–6].” It is important to state that dislocations play an important role on the high-pressure stability of metals an alloys; citing D Smith et al. 2017 J. Phys.: Condens. Matter 29 155401 and D Errandonea et al. 2018 J. Phys.: Condens. Matter 30 295402.
- 5 looks as rotated, not well aligned.
- 7 has a low resolution.
- Be sure all acronyms have been defined.
- They stacking fault mechanism described in page 12 could explain the fcc-hcp martensitic transition observed in rare-gas solids; D. Errandonea et al. Phys. Rev. B 65, 214110 (2002) and H. Cynn et al. Phys. Rev. Lett. 86, 4552 (2001).
- Increase the size of Fig. 29. Now is difficult to read letters.
Author Response
Ref. No.: materials-1064961
Title: Interactions of Dislocations and Boundaries during Deformation
Materials
Dear Editor,
Thank you very much for your attention and the reviewer’s comments concerning our manuscript (Ref. No.: materials-1064961). These comments are all valuable and very helpful for revising and improving our paper. We have studied these comments carefully and made a careful correction in the revised manuscript (The revisions are outlined by blue text). The detailed responses to the reviewer's comments are as follows.
Reviewer's comments:
Question #1
At the beginning of the introduction “The mechanical properties of metals at ambient temperature are mainly affected by their microstructures, except for the intrinsic properties determined by their chemical compositions [1,2]. This is because plastic deformation of metals is generated by dislocation activities, including nucleation and gliding of full edge/screw dislocations as well as nucleation and propagation of deformation twins [3–6].” It is important to state that dislocations play an important role on the high-pressure stability of metals and alloys; citing D Smith et al. 2017 J. Phys.: Condens. Matter 29 155401 and D Errandonea et al. 2018 J. Phys.: Condens. Matter 30 295402.
Answer: Thanks very much. The suggested papers are no doubt extraordinary works verifying the importance of dislocations on the stability of metals and alloys at high-pressure. It is our honor to read such works and we have learned a lot. After our discussion, we think these works are not so relevant to our topic on interactions between dislocations and boundaries. I am sorry that these works are not cited in the introduction of revised manuscript. We hope for your understanding about it.
Question #2
5 looks as rotated, not well aligned.
Answer: Thanks for your helpful suggestion. We have revised the Figure 3 to make it aligned. The revision is on the Page 6.
Question #3
7 has a low resolution.
Answer: Thank you for your helpful suggestion. We have improved the resolution of Figure 5. The revision is on the Page 8.
Question #4
Be sure all acronyms have been defined.
Answer: Thanks for your careful comment. We have supplemented and revised the following descriptions: face-centered cubic (fcc), body-centered cubic (bcc), hexagonal close-packed (hcp) and stacking fault energy (SFE). The revisions are on the Page 2, 7, 8 and 11.
Question #5
The stacking fault mechanism described in page 12 could explain the fcc-hcp martensitic transition observed in rare-gas solids; D. Errandonea et al. Phys. Rev. B 65, 214110 (2002) and H. Cynn et al. Phys. Rev. Lett. 86, 4552 (2001).
Answer: Thanks for your helpful suggestion. We have cited the important references to explain the stacking fault mechanism. Please refer to the blue text on the Page 12.
Question #6
Increase the size of Fig. 29. Now is difficult to read letters.
Answer: Thanks for your careful comment. We have increased the size of Figure 27. The revision is on the Page 37.
Thanks again for your careful comments. We hope you would be satisfied with this revision!

Round 2
Reviewer 2 Report
Now, the manuscript can be accepted.